

# ESD Reviews: Thermodynamic optimality in Earth sciences. The missing constraints in modeling Earth system dynamics?

Martijn Westhoff[1], Axel Kleidon[2], Stan Schymanski[3], Benjamin Dewals[4], Femke Nijsse[5], Maik Renner[2], Henk Dijkstra[6], Hisashi Ozawa[7], Hubert Savenije[8], Han Dolman[1], Antoon Meesters[1], and Erwin Zehe[9]

[1]Vrije Universiteit, Amsterdam, The Netherlands
[2]Max-Planck-Institut für Biogeochemie, Jena, Germany
[3]Luxembourg Institute of Science and Technology, Esch-sur-Alzette, Luxembourg
[4]University of Liege, Liege, Belgium
[5]University of Exeter, Exeter, UK
[6]Center for Complex Systems Studies, Department of Physics, Utrecht University, Utrecht, The Netherlands
[7]Hiroshima University, Hiroshima, Japan
[8]Delft University of Technology, Delft, The Netherlands
[9]Karlsruhe Institute of Technology (KIT), Karlsruhe, Germany

**Correspondence:** Martijn Westhoff (m.c.westhoff@vu.nl)

**Abstract.**

Thermodynamic optimality principles have been often used in Earth sciences to estimate model parameters or fluxes. Applications range from optimizing atmospheric meridional heat fluxes to sediment transport and from optimizing spatial flow patterns to dispersion coefficients for fresh and salt water mixing. However, it is not always clear what has to be optimized and how. In this paper we aimed to clarify terminology used in the literature and to infer how these principles have been used and when they give proper predictions of observed fluxes and states.

We distinguish roughly four classes of applications: predictions using a flux-gradient feedback, predictions using a constant thermodynamic potential boundary conditions, predictions based on information theoretical approaches and comparative studies quantifying entropy production rates from observations at different sites. Here we mainly focus on the flux-gradient feedback, since it results in clear physical limits of energy conversion rates occurring in the Earth system and its subsystems. We show that within the flux-gradient feedback application, maximum entropy production is in many cases equivalent to maximum power and maximum energy dissipation. We advocate the maximum power principle above the more widely used maximum entropy production principle because entropy can be produced by all kinds of fluxes, but only optimized fluxes performing work coincided with observations. Furthermore, the maximum power principle links to the maximum amount of free energy that can be converted into another form of energy. This clearly separates the well defined physical conversion limit from the hypothesis that a system evolves to that limit of maximum power. Although attempts have been made to fundamentally explain why a system would evolve to such a maximum in power, there is still no consensus. Nevertheless, we think that when the maximum power approach is correctly and consistently used, the positive (or negative) results will speak for themselves.

We end this review with some open research questions that may guide further research in this area.

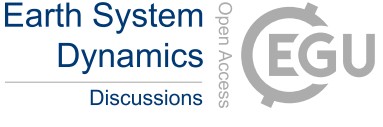

*Copyright statement.* TEXT

# 1 Introduction

Newtonian physics is the basis for many descriptions of phenomena in Earth sciences. Basis for these models are the energy, momentum and mass balances. They are widely and successfully employed in many different applications.

Most phenomena in Earth sciences are governed by the equation of classical mechanics and thermodynamics; modeled using energy, momentum and mass balances. Even though these equations are well-understood, often semi-empirical equations or constants are used. These are mainly used to model processes in the interfaces where particularly complex processes take place. Examples of such empirical descriptions are e.g. the Darcy equation for groundwater flow, the Chezy-Manning-Strickler formulation for friction in open channel flow, the Penman-Monteith equation describing evaporation, or the Von Kármán

constant to describe friction at interfaces. These phenomena are so complex that empirical parameters are generally seen as unavoidable.

    Nearly all environmental processes are subject to dissipative energy losses for instance due to friction. Frictional dissipation limits, due to the associated production of entropy, the rate of free energy that can be converted into another form of free energy. For example, if an object is falling freely, the kinetic energy increases at the expense of potential energy. In the absence

of friction, the mass keeps accelerating. However, in the presence of friction, the acceleration of the object reduces, ultimately, towards zero. In the final steady state the remaining potential energy is no longer converted into kinetic energy, but converted into heat instead. The asset is that thermodynamics provides fundamental limits to such conversions. And such limits imply optima (either minima or maxima) in system states. The hypothesis of thermodynamic optimality principles (TOPs) is that a system evolves to a state near such an extremum (either minimum or maximum) and remains there as long as its boundary

conditions remain stable.

    The optima in the TOP hypotheses arise often from the feedback between an thermodynamic potential gradient driving a flux and the flux depleting that gradient (see section 2). This means that such feedbacks should be explicitly described for a system to be able to determine the thermodynamic limits. Currently boundary conditions are often characterized by prescribed states, which are not necessarily independent from dynamics in the system. Naturally such an approach cannot account for the

flux gradient feedback. So the issue is to clearly identify the nature of the boundary conditions and how the system interacts with these.

    The idea of thermodynamic optimality principles is not new. The maximum power (MP) principle was proposed already by Lotka (1922) for biological systems and extended to ecosystems by Odum and Pinkerton (1955). Within atmospheric sciences, Paltridge (1975, 1978) formulated the related maximum entropy production (MEP) principle to explain the observed

atmospheric Equator to Pole heat flux, with their corresponding temperature profile. Interestingly, in 2005, he stated that he "began [with a] more-or-less random search for an extremum principle", without a proper understanding of the physics behind it (Paltridge, 2005). In fact TOPs have been applied in to a range of different systems in an ad-hoc manner, without a priori understanding about their applicability (see section 3 and 4)



The promises of TOPs are 1) a great potential to reduce the need for empirical parametrization or model calibration; 2) a shortcut to the right answer, by yielding effective macroscale parametrization (especially useful when there is a lack of observations; for example in the case of exoplanets or paleoclimate systems) 3) a reduced need for measurements and 4) it may constrain system aging and/or adaption of system functioning under change. But despite some very successful applications, there is also a lot of skepticism, mainly related to the lack of theory explaining why certain systems should evolve to a state of maximum entropy production for a particular component of their energy transfer (e.g. Goody, 2007; Volk and Pauluis, 2010; Ross et al., 2012). We argue that this skepticism is fueled by 1) inconsistencies in the use of thermodynamic concepts and terminology; 2) the lack of a theory postulating a priori the respective ranges of applicability of the different TOPs Martyushev and Seleznev (2014) identified this issue as one of the main reasons for criticism on TOPs – and listed several restrictions of the maximum entropy production principle; 3) apparently arbitrary choices of processes and boundaries considered in systems with many interlinked processes producing power and entropy (Volk and Pauluis, 2010). 4) a lack of a widely accepted physical foundation that explains why a system should evolve to a maximum/minimum; and 5) claims that the MEP principle is an inference principle, making it a hypothesis that cannot be rejected (because you can always argue that a rejection is caused by a missing constraint: see e.g. Ross et al., 2012).

In this paper, we aim to review the progress and applications of TOPs in Earth sciences. Special emphasis is given to:

1. Clarifying terminology and different principles used in literature

2. Better understanding on when these principles may be valid

3. Better understanding on how to apply these principles

Many publications have discussed TOPs over the last 15 years, including sevelar review papers (e.g. Ozawa et al., 2003; Martyushev and Seleznev, 2006; Kleidon and Schymanski, 2008; Quijano and Lin, 2014), special issues (Kleidon et al., 2010), or books (Kleidon and Lorenz, 2005; Dewar et al., 2014; Kleidon, 2016) have been published over the last ∼15 years. Here we focus on applications to (subsets of) the Earth system and distinguish between two main conceptual system characterizations: those incorporating an explicit flux-gradient feedback (Section 3) and those not considering such feedbacks (Section 4). Our aim hereby is to filter out the most fundamental common ground between successful application of TOPs and identify the most promising way toward a more unified theory.

## 2 From thermodynamic limits to thermodynamic optimality principles

In this section we shortly illustrate the principles behind TOPs. We will start with explaining the well-established and widely accepted Carnot limit. Then we discuss the effect of boundary conditions and how interactions between fluxes and boundary conditions lead to the maximum entropy production (MEP) principle and the more recently proposed maximum power (MP) principle. This is followed by a short discussion on TOPs as an inference principles.



## 2.1 The Carnot limit

The basis of TOPs is the first and second law of thermodynamics. The first law deals with the conservation of energy, while the second law states that entropy of an isolated system cannot be reduced, as it is produced by irreversible processes, while reversible processes (which can transport entropy between sub-systems) neither produce nor destroy entropy.

For a system of two connected reservoirs of different, but constant temperature ($T_h$ and $T_c$, for the hot and cold reservoir, respectively), the temperature difference initiates a heat flux $J$ flowing from the warm to the cold reservoir (Fig. 1a). For this system, the internal entropy production rate $\sigma$ (which in steady state balances the rate of entropy exported out of the system) is given by (e.g. Kleidon, 2012, Eq. 2.4):

$$\sigma = \left( \frac{J_{out}}{T_c} - \frac{J_{in}}{T_h} \right) \geq 0 \tag{1}$$

where the subscript $in$ and $out$ define the heat flux entering (and leaving) the hot reservoir and the heat flux leaving (and entering) the cold reservoir, respectively. Note that in steady state $J_{in} = J_{out}$.

    However, if the engine performs work – with $G$ being the rate of mechanical work – this is subtracted from the heat flux ($J_{out} = J_{in} - G$, Fig. 1b). Based on both laws of thermodynamics, it can be shown that the maximum rate of work ($G$) such an engine can produce (implying no entropy production: $\sigma = 0$) is given by the product of the heat flux leaving the warm reservoir

and the ratio of the temperature difference over the temperature of the warm reservoir. This maximum rate of work reflects the maximum generation rate of free energy and is defined as the Carnot limit. It is given by:

$$G_{carnot} = J_{in} \frac{T_h - T_c}{T_h} \tag{2}$$

with $J_{in}$ being the energy flux added to the hot reservoir to maintain its temperature, which is equal to the heat flux leaving the hot reservoir, and is partly used to perform work.

## 2.2 Flux-gradient feedback, maximum power limit, dissipation and MEP

The Carnot limit is derived for a system with a fixed temperature difference. This means there is no interaction between the boundary conditions and the heat flux. However, in many Earth systems, this flux-gradient feedback is established. For example, heat being exchanged between Equator and Poles decreases the temperature difference between the two. The same is true for vertical heat exchange between the Earth surface and the top of the atmospheric boundary layer. This implies that

for the boundary condition an incoming heat flux should be taken instead of the fixed temperatures of both reservoirs (Fig. 1c-f). In the above mentioned examples, this incoming heat flux is the net absorbed solar radiation. Spatial differences in the amount of solar radiation (e.g. at the Equator and Poles or at the soil surface and the top of the atmospheric boundary layer) cause, and maintains a temperature difference. But it also depends on the heat exchange between the two locations how large this temperature difference is.



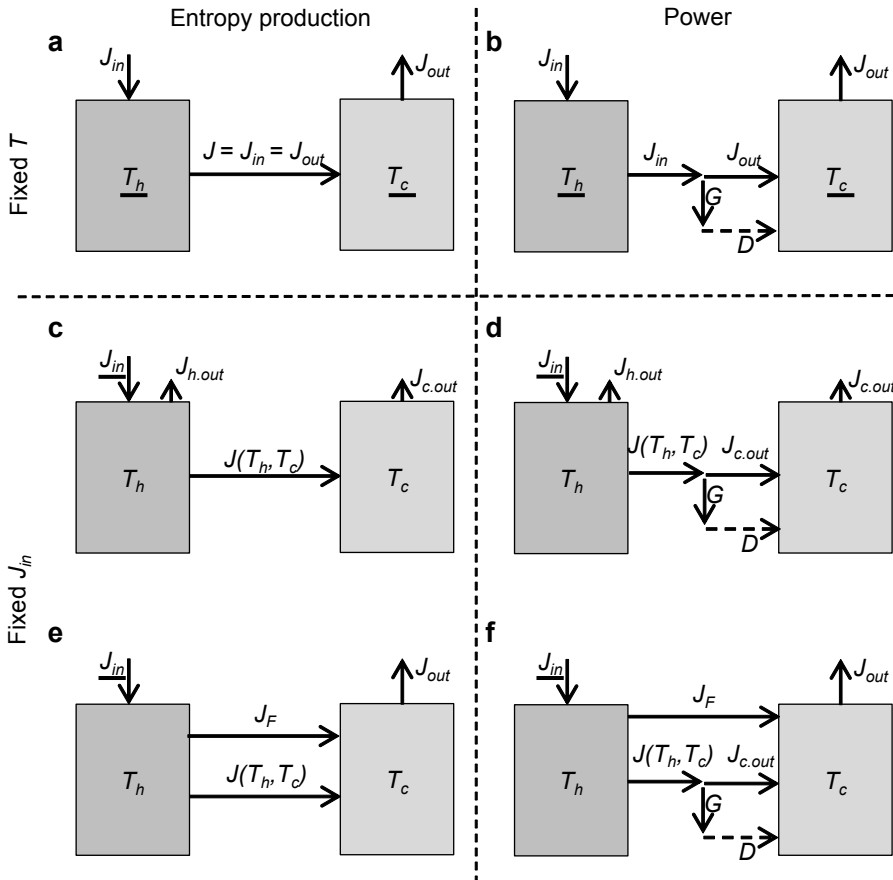

**Figure 1.** Schematization of a-b) a system with a fixed temperature difference without (a) and with (b) power subtracted; c-d) a system with fixed heat input ($J_{in}$) and leaking flux ($J_{h.out}$) without (c) and with (d) power subtracted; e-f) a system with fixed heat input ($J_{in}$) and leaking flux ($J_F$) without (c) and with (d) power subtracted. The heat being released by the dissipated power ($D$) feeds back on either the hot reservoir ($T_h$) or the cold reservoir ($T_c$), $J$ reflects heat fluxes, $G$ the subtracted power and $T$ the temperature. The subscripts $c$ and $h$ stand for hot and cold and $in$ and $out$ stand for incoming and outgoing. The underlined parameters denote the fixed boundary condition.



So if we relax the assumption of a fixed temperature difference, but assume a fixed heat supply instead, and still assume a steady state, then entropy production is given by:

$$\sigma = J\left(\frac{1}{T_c} - \frac{1}{T_h}\right) \geq 0 \tag{3}$$

where $J$ is the heat flux leaving the hot reservoir for which counts that in steady state $0 \leq J \leq J_{in}$. Note that $J$ is driven by the temperature difference $T_h - T_c$. In cases where $J < J_{in}$, an extra outgoing energy flux is needed to close the energy balance of the hot reservoir: in Fig. 1c-d, this is the (radiative) flux $J_{h.out}$, but it could also be an extra energy flux between both reservoirs as in Fig. 1e-f. This extra energy flux must have a known relation with the temperature (difference) and is called 'leakage' (e.g. Odum and Pinkerton, 1955) or a 'competing flux' (e.g. Westhoff et al., 2014).

In the extreme case where $J = 0$, the entropy produced by $J$ ($\sigma$) will be zero. In the other extreme where $J = J_{in}$ (in case the flux resistance goes to zero), the temperature of the hot reservoir will go down until it equals the temperature of the cold reservoir, leading to a zero entropy production by the flux $J$ as well. In-between these two extremes $\sigma$ is larger than zero, implying the existence of a maximum. If the flux $J$ is used to perform power, then power can be derived in a similar way as the Carnot limit and is then given by:

$$G = J\frac{T_h(J) - T_c(J)}{T_h(J)} \tag{4}$$

which, depending on the extend of the flux-gradient feedback, also bears a maximum for a certain value of $J$.

The Carnot limit assumes that the generated work is used and dissipated outside the system. However, in Earth systems it generally does dissipate within the system. And if all power is being dissipated within the system, then MP corresponds to maximum energy dissipation. This has resulted in other maximization terms such as maximum kinetic energy dissipation (MKED) or maximum free energy dissipation (see section 3), depending on what form of energy is being dissipated.

It is interesting to see that in the derivation of power (Eq. 4) entropy production is assumed to be zero, which seems to contradict MEP. However, one has to keep in mind that when all generated power is being dissipated within the system again, the steady state temperature $T_c$ will be the same as in the case where no power has been generated (and $\sigma > 0$). So, one can say that although MEP and MP are equivalent to each other, MEP ignores the potential cascade of energy conversions (e.g. from potential to kinetic to electrical energy) before all generated power will be dissipated. Thus, MP can also be used to predict limits of such conversion rates happening in other Earth system processes (see section 2.3).

Besides that, care should be taken when using MEP, because there are other fluxes producing entropy, that do not dissipate mechanical energy. An example of this is radiative fluxes that may reduce temperature differences (e.g. flux $J_F$ in Fig. 1e-f), but that do not perform any work. For this reason, (Ozawa et al., 2003) made a clear distinction between entropy production due to thermal dissipation (caused by a flux dissipating the temperature gradient without performing work) and entropy production due to mechanical or viscous dissipation (caused by a flux that does perform work). The latter has, similar to power, to do with motion and transport of energy or mass towards a location of lower chemical potential. Another aspect is that MP separates the thermodynamic limit from the hypothesis that a system evolves to that limit (Kleidon, 2016, p. 95), while MEP implicitly combines both. This separation is advantageous, since the thermodynamic limits are very well established, while the hypothesis that a system evolves to and operates at such a limit may be true for some systems or processes, but not for others.

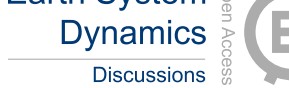



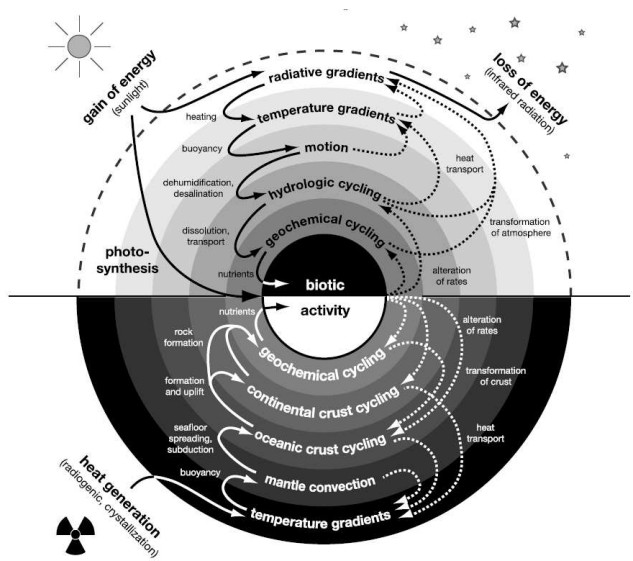

**Figure 2.** Simplified summary of a hierarchy of power transfer among Earth system processes. Solid arrows describe flows of energy, while dotted arrows describe their effects (taken from Kleidon, 2010)

## 2.3 TOP beyond heat

So far we have shown that a temperature difference can be used to create free energy, which is ultimately being dissipated again. But before being dissipated completely, several free energy conversions are possible. A simplified overview of the hierarchy of energy conversions in the Earth system is illustrated in Fig. 2. For example: spatial differences in solar radiation

result in a temperature gradient, which cause convection cells. And these convection cells also transport water vapor through the atmosphere, resulting in turbulent heat fluxes (represented by the hydrological cycle). Terrestrial precipitation results in potential energy differences of water at the Earth surface, resulting in kinetic energy transporting water to points of lower potential energy; the flowing water can transport sediment resulting in erosion, depleting the potential energy gradient of the river bed. A similar hierarchy is also illustrated for the Earth's internal processes. Several of these conversions we will consider

in section 3.

     Important to notice is that all these different forms of free energy are driven by energy potential differences, while the resulting fluxes aim to reduce these differences. Thus, similar as described above, there is likely a theoretical maximum in power that is transferred by the different fluxes. Another important thing to notice here is that energy conversion means that one form of energy transforms into another, meaning that we have to account for this in the energy balance of both forms of

free energy. An issue that is often not taken into account explicitly.



### 2.4 TOPs as inference principle

It has been argued that MEP is not a physical but rather an inference principle, (e.g. Goody, 2007; Dewar, 2009). This implies that it is just a way to obtain macroscopic properties of a system, while neglecting small scale motion: For example, temperature and pressure are effective properties of the kinetic energy of all molecules of a gas, while a diffusion coefficient can be seen as a macroscopic average of the interaction of all molecules.

Dewar (2009) showed that the maximum entropy (MaxEnt) formalism, which has its origin in information theory (which he hinted to be the basis for MEP[1]), is just a way to see if the system is described by sufficient physical assumption and constraints – which he called the "essential physics". It then depends on the system of interest which physics are essential. With this respect, he argued that, given the close correspondence with observation, the simple atmospheric model of Paltridge (1978, see section 3.1), apparently covers the essential physics, while all other physical details appear to be irrelevant at that scale.

At the same time he also argued that a bottom-up approach would in principle lead to the same results (an argument that was also made by Goody, 2007), albeit on the costs of more computational power while potentially running into problems when parameterizing sub-grid phenomena.

The fact that MEP can be seen as an inference principle, means that the MEP hypothesis cannot be falsifiable: with this respect Dyke and Kleidon (2010) stated that a mismatch with observations "should not be seen as a failure of MEP per se, but rather as the lack of further relevant information, for instance, in terms of additional constraints". Ross et al. (2012) used this argument (among others!) to conclude that MEP is not a scientific theory, and that "any predictions based on MEP should not be considered scientifically founded". However, with this conclusion he neglects the gain of (computational) efficiency pointed out by Dewar (2009).

Note, that although the discussion about inference has been entirely about MEP, it may very well be that it also counts for MP, albeit that within the MP hypothesis the well established limits on energy conversions is separated from the hypothesis that a system evolves to such a limit. Furthermore, we would like to stress that even if TOPs are 'just' inference principles, they may still be useful in predicting a range of fluxes and states (as we will see in section 3).

### 3 Applications with a flux-gradient feedback

In this section, we shortly review a range of relative recent studies applying TOPs to components of the Earth system. Our focus is on applications that have incorporated a flux-gradient feedback. A short summary is also given in Table 1.

### 3.1 Temperature gradient as driving force

Several successful studies have used (small variations of) the model of Fig. 1c. They used the MEP principle (using Eq. 1) to obtain vertical or horizontal turbulent heat fluxes (i.e. latent and sensible heat fluxes) within the atmosphere. The small

---

[1]In his paper, Dewar (2009) indicated that attempts to derive MEP form the MaxEnt formalism (Dewar, 2003, 2005) had some technical limitation, while he also hinted to an upcoming derivation. However, this derivation has not been published so far.



variations mainly consisted of adding more reservoirs in series to increase the spatial resolution. In these cases, the energy balance for each reservoir was solved, while maximizing the summed entropy production by all inter-reservoir heat fluxes.

Paltridge (1975, 1978) was the first to idealize the global atmosphere as 10 connected boxes from Pole to Equator to Pole. In the first paper (Paltridge, 1975), he did not maximize entropy production, but he minimized the average ratio of absorbed
solar energy to the longwave emission to space. This minimization led to a distribution of surface temperature and cloud cover that compares very well to observations. Only then, he showed that minimizing this ratio can equivalently be interpreted as a maximum in entropy production produced by the horizontal heat flux between Equator and Poles; a point he showed more consistently in Paltridge (1978).

Lorenz et al. (2001) applied basically the same approach to a two-box model and showed that MEP also correctly predicts
horizontal atmospheric heat fluxes on Titan and Mars. In fact they did not optimize $J$ itself, but rather an effective heat diffusion coefficient. This approach was criticized by Goody (2007) because the simple 2-box model did not account for the important role of the planetary rotation rate. To investigate this criticism, Jupp and Cox (2010) added a simple representation of the rotation rate to this 2-box model. In fact, they they proposed a way to assess if an atmosphere has sufficient degrees of freedom to be able to adapt to a state of MEP. Or in other words: If there are physical constraints preventing the system to adapt its
effective conductance (or diffusivity), the system can simply not evolve to a state in which it produces maximum entropy. They concluded that Earth, Titan and Mars all had atmospheres capable of adapting to the MEP state described in Lorenz et al. (2001), but also that some planets may not be able to do so. Similar results were found by Fukumura and Ozawa (2014) using multiple box models.

In addition to these setups, Pascale et al. (2012b) added a vertical component to the two-box model: the two horizontal
boxes were both vertically connected with a surface box with turbulent heat fluxes in-between (making it a combination of the models in Fig. 1c and e). Their MEP derived results compared well to results of a GCM. When the resolution in both vertical and horizontal direction was increased, the optimized meridional flux still compared rather well, but the vertical turbulent fluxes did not. The authors suggested that this is probably due to missing physical constraints: in their case the emitted longwave radiation of each vertical grid cell.

Interestingly, they also showed that the MEP state of the meridional flux is rather independent of entropy production of the vertical turbulent fluxes. This is an important finding since other studies aimed to optimize vertical turbulent heat fluxes without considering the meridional heat fluxes: Ozawa and Ohmura (1997) divided the vertical atmospheric column into a number of grid cells and solved the energy balance for each grid cell taking only shortwave radiation, longwave radiation and a convective flux into account, while entropy production of the convective flux was maximized (comparable to the model in Fig. 1e). The
obtained fluxes deviated a little from observed ones, which the authors attribute to the assumption of a gray atmosphere. This motivated Herbert et al. (2013) to extend the model with more realistic radiative properties. This led to more realistic shapes of the temperature profiles, but a quantitative comparison with observations is missing.





**Table 1.** Overview of studies applying TOPs to Earth system dynamics with a flux-gradient feedback

| Author(s) | TOP | model | degree(s) of freedom being optimized |
|---|---|---|---|
| **Temperature gradient as driving force** | | | |
| Paltridge (1975) | min. ratio of absorbed solar energy to outgoing longwave radiation | ∼Fig. 1c | surface temperature, cloud cover and meridional heat flux |
| Paltridge (1978) | MEP | ∼Fig. 1c | surface temperature, cloud cover and meridional heat flux |
| Lorenz et al. (2001) | MEP | Fig. 1c | heat diffusion coefficient |
| Jupp and Cox (2010) | MEP | Fig. 1c | drag coefficient being related to surface heat diffusion coefficient (MEP only works when system has sufficient abilities to adapt) |
| Fukumura and Ozawa (2014) | MEP | ∼Fig. 1c | meridional heat flux (MEP only works when system has sufficient abilities to adapt) |
| Pascale et al. (2012b) | MEP | ∼Fig. 1c,e | vertical turbulent heat fluxes and meridional heat flux |
| Ozawa and Ohmura (1997) | MEP | ∼Fig. 1e | vertical convective flux |
| Herbert et al. (2013) | MEP | ∼Fig. 1e | vertical convective flux |
| Dyke et al. (2011) | MEP | Fig. 1e | Nusselt number, diffusion parameter |
| Kleidon and Renner (2013a) | MP | ∼Fig. 1f | effective exchange velocity |
| **Tops in GCMs** | | | |
| Kleidon et al. (2003) | MEP | GCM | friction parameter |
| Kleidon et al. (2006) | MEP | GCM | Von Kármán constant |
| Kunz et al. (2008) | MEP/MKED | GCM | friction and heating timescales, diabatic heating factor |
| Pascale et al. (2012a) | MKED | GCM | 8 empirical model parameters (only convective entrainment rate and cloud-to-droplet conversion rate could be constrained – MEP for sum of fluxes did not work) |
| **Hydrological models with water potential as driving force** | | | |
| Porada et al. (2011) | MEP | Fig. 3 | conductance for rootwater uptake and runoff (comparison with data is not convincing (factor 2 to 5 off)) |
| Westhoff and Zehe (2013) | MEP | ∼Fig. 3 | all empirical model parameters (only conductances and a non-linearity factor resulted in a mathematical maximum (MEP simulation did not match observations)) |



*continued from previous page*

| Author(s) | TOP | model | degree(s) of freedom being optimized |
|---|---|---|---|
| Westhoff et al. (2016) | MP | Fig. 3 | conductance for evaporation (strong assumptions on relations) |
| Wang et al. (2015) | MEP | Fig. 3 | conductance for evaporation and runoff (summed EP of both evaporation and runoff; strong assumptions on relations) |
| Zehe et al. (2010, 2013) | max. free energy diss. | 2D vertical | macropore density |
| **Spatial organization** | | | |
| Leopold and Langbein (1962) | MEP: local min. energy diss. | 2D random walk | drainage network configuration (Statistical Entropy definition is imprecise) |
| Rinaldo et al. (1992); Rodriguez-Iturbe et al. (1992a, b), | min. energy diss. | 2D | drainage network configuration |
| Hergarten et al. (2014) | min. energy diss. | 2D | drainage network configuration |
| Errera and Bejan (1998) | min. energy diss. | 2D | preferential flow network |
| Schymanski et al. (2010) | MEP | 2D | spatial organization of vegetation |
| **Other driving forces** | | | |
| Dyke et al. (2011) | MEP | | erosion constant |
| Kleidon et al. (2013) | MP | | sediment drag force, river density , erosion rate |
| Miller et al. (2011, 2015); Gans et al. (2012); Miller and Kleidon (2016) | MP | | drag coefficient |
| Zhang and Savenije (2018) | MP | | dispersion coefficient for mixing of salt and fresh water |

A similar model has been used to predict mantle convection in the Earth's interior and oceanic crust recycling (Dyke et al., 2011). For both processes, they used the conceptualization of Fig. 1e. In the case of mantle convection, $J_F$ and $J$ represented the conductive and convective heat flux through the mantle, respectively. For the case of oceanic crust recycling, $J_F$ represented

5  the conductive heat flux through the continental crust and $J$ represented bulk transport of heat through the oceanic crust. This latter flux was assumed to be a linear function of the temperature gradient in the oceanic crust, and to obtain the temperature field in the space-time domain, they stated that at the oceanic ridge (where hot mantle material rises to the surface), the temperature is high, after which it cools over time, which, in steady state, is a surrogate for distance from the ridge. In this way they introduced a creep velocity driven by a temperature gradient, which was subsequently optimized by MEP.

10  While all of the above mentioned studies used MEP to optimize the turbulent fluxes, Kleidon and Renner (2013a) used MP to be able to split the turbulent heat flux into sensible and latent heat fluxes. They used a model similar to Fig. 1f, where $J_F$ represented longwave radiation and $G$ the power generated by the sensible heat flux to transport water vapor (which is

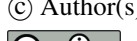



equivalent to latent heat) from the surface to the top of the atmospheric boundary layer. In follow-up papers, they applied this approach to test the sensitivity of the global hydrological cycle to changes in greenhouse and solar forcing (Kleidon and Renner, 2013b); to test the geographical distribution of sensible and latent heat (Kleidon et al., 2014) and to test the climate sensitivity of the land-ocean contrast (Kleidon and Renner, 2017).

## 3.2 TOPs in general circulation models

Moving away from the simple box models discussed in section 3.1, TOPS have also been applied to estimate parameters in general circulation models (GCM). With this respect, Kleidon et al. (2003) identified for their simplified GCM two cases needing different treatment: if degrees of freedom (e.g. wind circulation or turbulence) are explicitly simulated, a sufficiently fine resolution is needed to reach the MEP state. If this is not the case, the model parameterization can be tuned by MEP, which they did for an effective friction parameter controlling boundary layer turbulence.

In a follow-up study, Kleidon et al. (2006) extended this approach by adding turbulent exchange at the surface. They optimized the Von Kármán constant and found that the empirical value of 0.4 results in a state of MEP, which they showed to be equal to maximum kinetic energy dissipation (MKED).

In a similar approach, Kunz et al. (2008) used MEP and MKED as objective functions. They showed that parameters influencing boundary conditions cannot be optimized, which is in line with all the literature discussed above. Internal parameters influencing the large scale heat transport show maxima in both objective functions with optimum values being only slightly different. Pascale et al. (2012a) would later argue that these similar optima are caused by the fact that the water cycle, where evaporation and transpiration produce a lot of entropy, was ignored in the study of Kunz et al. (2008).

Pascale et al. (2012a) used a complex GCM including the water cycle and aimed to optimize eight empirical parameters. While material entropy production (entropy produced by latent and sensible heat) and planetary entropy production (material plus radiation entropy) did not show a maximum for any of the parameters (which is in line with the conclusions of Fraedrich and Lunkeit, 2008), a maximum in kinetic energy dissipation was found for realistic values of two out of eight parameters (convective entrainment rate and cloud-to-droplet conversion rate). The reason that MKED did show maxima is that the amounts of entropy production involved in evaporation and condensation are much larger than those produced by sensible heat fluxes, while only the latter can be optimized. If the summed entropy production of both processes are taken, the maximum vanishes due to the relative large fraction of entropy produced by the latent heat flux. Unfortunately, no good explanation was given why some parameters did and others did not converge to an optimum.

These studies show that it is not always clear which entropy production rates can be maximized. However, kinetic energy dissipation (which, in steady state, is the same as mechanical power generation) excludes many of the fluxes that do produce entropy but cannot be optimized (as discussed in section 2.2).

## 3.3 Hydrological models with water potential as driving force

Instead of temperature differences as driving force, Kleidon and Schymanski (2008) demonstrated the MEP principle conceptually with water potential as driving force. They used an electric circuit analogue of the terrestrial water balance in which



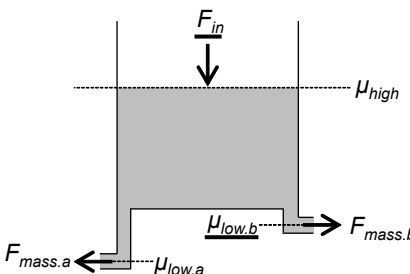

**Figure 3.** Analogue to the terrestial water balance model of Kleidon and Schymanski (2008). $F$ is a mass flux and $\mu$ a water potential.

evaporation competed with runoff for the available water, where both fluxes were described as a (in their case linear) function of the water potential. This model can be interpreted as a 1-box model as in Fig. 3. They showed mathematically that if one conductance was supposed to be known, the other one could be estimated with MEP. They defined entropy production as:

$$\sigma = F_{mass} \frac{\mu_{high} - \mu_{low}}{T} \tag{5}$$

Where $F_{mass}$ is the mass flux, $\mu_{high} - \mu_{low}$ is the potential difference driving the mass flux and $T$ is the temperature at which the process occurs.

Porada et al. (2011) applied this model to simulate the hydrological response of the 35 largest catchments of the world. They used an interesting way to optimize both, the conductance for baseflow ($c_{base}$) and the conductance for root water uptake ($c_{root}$, which is a surrogate for transpiration). They did this by first optimizing MEP by baseflow (finding the optimum value
for $c_{base}$ for a given $c_{root}$ value) for a whole range of $c_{root}$ values, while in a second step they fixed $c_{base}$, and optimized $c_{root}$ for MEP by root water uptake (Fig. 4). The combined MEP-state of baseflow and root water uptake was assumed to lie at the intersection of the two "ridges" of both optimizations. Comparison of the annual water balance of the 35 catchments was reasonable, but not convincing with errors in simulated mean runoff of up to 300% for temperate climates.

Nevertheless, this study inspired Westhoff and Zehe (2013) to apply a similar approach to a bucket model, which is very
close to the often used hydrological HBV model (Lindström et al., 1997). They describe entropy production in the same way as Kleidon and Schymanski (2008), but by assuming isothermal conditions, they 'transferred' it to the MP principle. They simply tried to optimize each of the 9 free model parameters and concluded that most parameters could not be constrained in a meaningful manner. However, they did show that besides resistances, also parameters that are only linear in the log-transform space could, at least mathematically, be optimized.

In a follow-up, Westhoff et al. (2016) returned to the 1-bucket model of Fig. 3. Instead of using a linear relation between the evaporative flux and its driving potentials, they derived a non-linear relation in such a manner that at MP, the splitting of effective rainfall into evaporation and runoff always lies somewhere on the asymptotes of the Budyko curve (an empirical curve relating the aridity index of a wide range of catchments to their evaporation fraction Budyko, 1974). By introducing



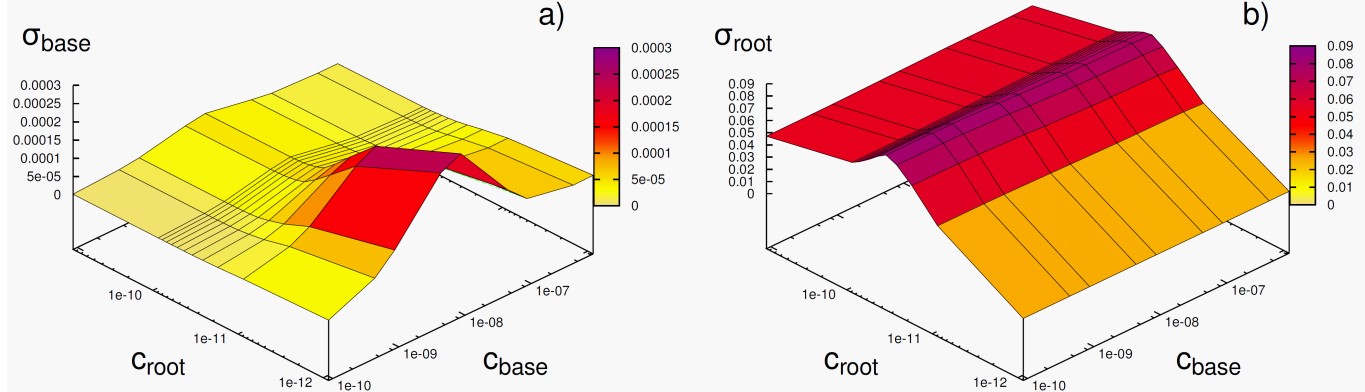

**Figure 4.** Entropy production of (a) baseflow and (b) root water uptake as a function of the two model parameters $c_{base}$ and $c_{root}$. The combined MEP-state of baseflow and root water uptake lies at the intersection of the two "ridges" in (a) and (b), (taken from Porada et al., 2011)

dynamics in forcing and actual evaporation (for which Westhoff et al., 2014, showed that this may lead to a second optimum), they showed that at MP the Budyko curve deviated from the asymptotes towards more realistic values. Although their results compare reasonable with observations, physically derived fluxes were absent. It is therefore unclear if their results are indeed a result of thermodynamic optimality, or if it is just a result of a mathematical exercise in which the mathematical relations

between fluxes and gradients have, implicitly, been optimized instead of the fluxes alone.

Similar questions can be asked about the approach of Wang et al. (2015) who also started with the model formulation of Kleidon and Schymanski (2008) and showed that the combined optimization of the evaporation and runoff transfer parameters provides a thermodynamic foundation of an empirical formulation for the Budyko curve which has been derived by the so-called proportionality hypothesis (Wang and Tang, 2014). The questionable aspects of this study concern the formulation of

the evaporation and runoff fluxes, which seem to lack the correct physics but are required to gain the desired result.

All these applications had deficits: they either did not correspond well to observations (Porada et al., 2011), or they lacked the correct physical description of the different fluxes (Westhoff and Zehe, 2013; Wang et al., 2015; Westhoff et al., 2016).

A completely different approach of TOPs was performed by Zehe et al. (2010, 2013), who used maximum free energy dissipation as objective function. This basically means that a system tends to go back to thermodynamic equilibrium (without

any gradient driving a flux) at the fastest possible rate. They also used the water potential differences as driving force: more precisely, they focussed on the interplay between capillary binding energy (or matric potential) and potential energy in the unsaturated zone. In the first study (Zehe et al., 2010) they showed that dead-ended macropores in the unsaturated zone of cohesive soils accelerate depletion of matric potential gradients during single rainfall events after long dry spells, which implies accelerated reduction of free energy of soil water. However, they did not find an optimum macropore density that maximized

free energy dissipation. In their follow-up study (Zehe et al., 2013), they extended the simulation period from 25 hours to 1.5 years. By varying the macropore density, they obtained two maxima in free energy dissipation when averaged over rainfall





driven conditions, of which the first one corresponded very closely to the calibrated density. Running the same model for a much wetter catchment with highly conductive soils did not reveal a maximum in free energy dissipation. This is caused by the fact that the macropores in that system enhance drainage, while the dead-ended pores in the cohesive soils enhanced wetting of the soil. They found, however, a distinguished macroporosity which brings the system closest to steady state in the

sense that average gains in potential energy of soil water during rainfall-driven conditions are equal to average export rates of potential energy by subsurface storm flow and groundwater recharge. This distinguished macroporosity was also close to the calibrated one. Intriguing in these studies is that macropores in the cohesive soils were created by earthworms, which immediately raises the question why these earthworms would dig exactly that amount of burrows that leads to a maximum in free energy dissipation. The authors hypothesized that this is because of the fact that, if the worms would behave differently,

they would work against the geomorphic resilience of their habitat (Zehe et al., 2010). It may thus be a selection criterion for earthworms. However, this hypothesis has, up to now, not been tested.

### 3.4   Spatial organization

TOPs have also been used to optimize spatial organization, for instance in terms of flow networks or vegetation patterns. In these applications, the parameter to optimize is not a specific flux or its conductance, but rather the location and connection

of different grid cells within an area or volume. In this section we discuss the organization of flow networks and spatial organization of vegetation.

   To infer the dendritic network often seen in flow networks such as rivers, a couple of, at first sight, different optimization principles have been applied. Leopold and Langbein (1962), were probably the first aiming to use entropy to infer on the structure of water flow paths. Their idea was to infer on the most probable state, where they aim to link Shannon (information)

entropy with thermodynamic entropy. They proposed that the most probable state of a steady state river configuration is one minimising entropy production and consequently minimizing work or energy expenditure. The most probable state is then the one where work is uniformly performed along the channel

   While the final outcome of their minimum energy expenditure principle is in line with later studies (Rinaldo et al., 1992; Rodriguez-Iturbe et al., 1992a, b; Hergarten et al., 2014), there seems to be a few weak points in their derivation: The first one

is that their definition of statistical entropy is flawed[2], while the second one is in their derivation of probability in which they replace the exponent of the Bolzmann distribution by an entropy production term. So it seems that they mix entropy production with entropy itself, making it unclear if it is indeed a thermodynamic optimality principle or an inference principle originating from information theory.

   But as said before, their interpretation of minimum energy expenditure is in line with later studies, albeit from a different

starting point. In a series of papers Rinaldo et al. (1992); Rodriguez-Iturbe et al. (1992a, b) developed the so-called Optimal Channel Network (OCN) from the hypothesis that a river network configuration develops such that frictional losses (or energy expenditure) are minimized. The rate of frictional losses were determined by the loss of potential energy over the course of a

---

[2]Leopold and Langbein (1962) defined entropy $S$ as $S = \sum \ln p_i$, while it should have been $S = \sum p_i \ln p_i$, with $p_i$ being the probability of a state with a given energy $E_i$





river stretch, constrained by the fact that the slope of the river is proportional to the flux to the power of $-\theta$, with $\theta \approx 0.5$. This constraint basically means that a large river has a small slope and a small river a large one. Keep in mind that the mass flux increases along the flowpath of the river, since it is assumed that each grid cell adds a unit mass flux (Rodriguez-Iturbe et al. (2011) showed that this latter assumption is indeed correct for, at least, the Upper Rio Salado basin in New Mexico, despite the

heterogeneous spatial distribution of rainfall over the basin). A minimum was found for dendritic networks having the same statistical properties as observed river networks obtained from digital elevation models. Hergarten et al. (2014) extended this approach to subsurface networks, with the constraint of a constant total porosity and a power law relation between hydraulic conductivity $K$ and porosity, leading to very similar results as for the OCNs.

Errera and Bejan (1998) used the same principle to let a network evolve over time. They showed within a 2D simulation

how fractal networks form when a grid cell, experiencing a shear stress higher than a critical one, is eroded away. Although not explicitly shown, the evolution of fractals stops when the network evolved so far that the shear stress is everywhere too low to erode any other grid cell.

Although the used principles in these applications are minimization principles, they are equivalent to the maximum free energy dissipation principle proposed by Zehe et al. (2010, 2013): Thermodynamic equilibrium (i.e. the absence of any driving

gradient) is reached if all water is at the level of the outflow (potential and kinetic energy are both zero). Maximum free energy dissipation thus means that all potential energy (and thus all water) should be exported out of the system as fast as possible. Within the OCN framework this means that friction should be minimal which is equivalent to minimum energy expenditure. Besides that, Kleidon et al. (2013) showed that minimization of dissipation within the flow network is accompanied by a maximization of dissipated power outside the network combined with maximum work on sediment transport (see section 3.5).

In a different approach, Schymanski et al. (2010) used the MEP principle to predict spatial organization of vegetation. They kept track of the product of all fluxes and gradients of a simple 2D deterministic model developed by Klausmeier (1999) leading to striped pattern formation and divided that by (a constant) temperature. They first showed that entropy production indeed increased with a higher degree of vegetation organization (due to changes in fluxes and gradients), with a maximum for a striped pattern. In a second step they reduced this model to a two box model in which they showed that the areal fraction of

vegetation leading to MEP is in close accordance with the 2D model. Although they incorporated the flux-gradient feedback, they maximized the summed entropy production terms of all fluxes. It is unclear if similar results would have been achieved if, instead of the summed entropy production terms, only the entropy production of the internal water fluxes (which would be in line with Ozawa et al., 2003; Pascale et al., 2012a) had been taken into account.

### 3.5   Other driving forces

Besides temperature and water potential gradients, there are other forces driving fluxes related to Earth sciences.

The first application we discuss here is the geopotential of the continental crust enhancing erosion (Dyke et al., 2011). Besides the mantle convection and oceanic crust recycling, as discussed in section 3.1, they also used MEP to predict global erosion rates. They used the feedback that a higher geopotential of the continent (which is pushed-up by buoyancy forces within the mantle) results in a higher erosion rate, while the erosion depletes the geopotential. They used Eq. 5, with $F_{mass}$



being the erosion rate and $\mu_{high} - \mu_{low}$ the geopotential between the top of the continental crust and the bottom of the sea floor where the sediments settle again. Their MEP derived erosion rate was about one order of magnitude higher than estimates of Syvitski et al. (2005).

The thermodynamic limit of the conversion of kinetic energy of water to that of sediments was qualitatively explored by
Kleidon et al. (2013). They used the feedback that kinetic energy of the water is used to lift and transport sediment, which reduces the kinetic energy of both, increasing sedimentation. This feedback yields a maximum in power to transport sediments. In a second step they demonstrated that the presence of channels leads to a reduction of frictional dissipation of water flow (which is in line with the optimal channel network approach described in section 3.4). So, the system can export the detached sediments at a faster rate. In the last step they showed that the feedback between continental uplift and the depletion of
this geopotential by sediment transport again has a maximum in uplifting power. Although they used a similar approach as Dyke et al. (2011), they explicitly focus on the power that is needed to lift the continental crust. These three combined steps show that a minimization of dissipation within the flow network is accompanied by a maximization of dissipated power outside the network.

An approach looking at the limits to convert kinetic to electrical energy, comes from the set of papers by Miller et al. (2011);
Gans et al. (2012); Miller et al. (2015); Miller and Kleidon (2016). They looked at electrical energy generated by wind turbines. The feedback they used was that a higher drag force by wind turbines (which uses this force to creates electrical energy) leads to a lower momentum (and thus less kinetic energy) of the wind. Especially for large wind farms, this feedback may lead to significant reductions of electrical power. Comparison with results from a GCM in which they implemented wind turbines yielded similar results.

The MP principle has also been applied to the mixing of fresh and saline water in estuaries (Zhang and Savenije, 2018). The potential energy provided by the river discharge is converted into work by lifting up and mixing the more saline deeper water with the fresher water near the surface. Maximizing the power with respect to the dispersion of saline into fresh water then led to a new expression for the dispersion, which is similar to earlier empirical relations. Although they retrieved excellent results, the incorporated feedback did not seem to be the correct one (personal communication of the original authors of the
manuscript): it is not the potential energy difference causing mixing, but an angular momentum originating from the fact that center point of the horizontal pressure caused by the heavier salt water is at a lower altitude than the centre point of that of the fresh water, while the pressure at both sides are equal. A larger radius for the angular momentum enhances mixing, reducing the density differences along the estuary and thus reducing the radius.

This last example shows that even if the optimized fluxes corresponds closely to empirical data, it is not necessarily a proof
that TOPs are used in the correct way or that they are even valid.

## 4   MEP applications without a flux-gradient feedback

Although our focus has been on flux-gradient feedbacks leading to maxima in thermodynamic entropy production, power or energy dissipation there are several application found in the literature which did not have this flux boundary condition. Since we



**Table 2.** Overview of studies applying MEP **without** a flux-gradient feedback

| Author(s) | Topic | Comments |
|---|---|---|
| **Constant driving gradient** | | |
| Ozawa et al. (2001) | turbulent fluid systems | MEP equals maximum flux propositions for turbulent flows |
| Jupp and Schultz (2004) | hydrothermal convection cells | MP equals temperature at bottom of convection cell |
| Börsing et al. (2017) | hydrothermal convection in a confined porous medium | MEP can predict number of convection cells |
| Konings et al. (2012) | atmospheric hydrological cycle | max. efficiency for derived residual fraction of vapor in the atmosphere does not correspond to observations. Application looked at Carnot efficiencies |
| Mölg (2015) | atmosphere-glacier system | it has a flaw in the entropy production formula |
| **Applications from information theory** | | |
| Wang and Bras (2009, 2010, 2011); Huang and Wang (2016) | partitioning of the land surface energy balance | MEP derived results matches observations |
| del Jesus et al. (2012) | spatial organization of vegetation | MEP is proportional max photosynthesis |
| **Observed entropy production of different sites** | | |
| Holdaway et al. (2010) | ecosystem development | developed ecosystems produce most total entropy |
| Lin (2015) | ecosystem development | developed ecosystems produce most total entropy |
| Quijano and Lin (2015) | ecosystem development | compared entropy production by different fluxes - rejects hypothesis |
| Brunsell et al. (2011) | ecosystem development | more vegetation in a land surface model produces more entropy |

think that such different ways of applying (mainly) MEP, adds to the blurred view on TOPs, we will discuss these applications here. For an overview, see Table 2.

## 4.1 Constant driving gradient

The first study applying a constant driving gradient we discuss here, is the approach described by Ozawa et al. (2001). They
simply stated that for a system with prescribed states as boundary condition (such as Fig. 1a), entropy production is maximum
if the flux is maximum, which also follows directly from Eq. 1 (assuming a fixed $T_h$ and $T_c$). To determine this maximum flux
they used other constraints than MEP. So although they hint explicitly to MEP it is important to notice that they did not use it
to determine or constraint any flux.

At first sight, fixed temperature boundaries were also applied by Jupp and Schultz (2004) to simulate, with a relative simple
model, the subseafloor hydrothermal cells. Their approach can be summarized by a 4-box model similar to the model of Fig.





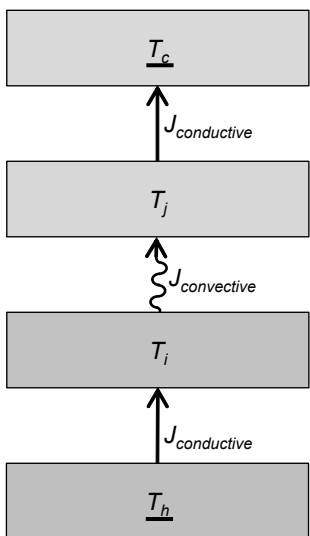

**Figure 5.** Analogue of the model used by Ozawa et al. (2001). $T_h$ and $T_c$ are the fixed temperatures, while $T_i$ and $T_j$ depend on the fluxes. Ozawa et al. (2001) assumes that the convective heat flux $J_{convective}$ is so efficient that $T_i \approx T_j$.

5 in which the first and last boxes have fixed temperatures representing the magma chamber and seafloor respectively. The middle boxes represent the temperature at the top of the impermeable layer (with only a conductive heat flux flowing through) above the magma chamber and the top of a reaction zone in which relative cold water from the sides is heated such that the convection cell evolves. Although the top and bottom boxes have been explicitly mentioned in the article, they merely served

as upper and lower limits of the temperatures in the middle boxes: The authors show that for a given value of the driving temperature (similar to $T_i$ in Fig. 5), the upwelling temperature (similar to $T_j$ in Fig. 5) corresponding to MP of the convective heat flux is $\approx 400°C$, which corresponds well to observations. Due to the non-linear behavior of water under high pressure and temperature, this optimized temperature appears to be rather insensitive to the driving temperature. So despite the fact that their figure 1 explicitly mentions the fixed temperatures at the boundary, they were not used in their final optimization procedure.

Instead they incorporated a flux-gradient feedback, by letting the upwelling temperature be a free parameter.

In the field of deep groundwater, MEP has been applied to explain the number of thermal convection cells in a confined porous medium (Börsing et al., 2017). With a high resolution 2D model, the convection cells were simulated explicitly, which is probably the reason why maxima in entropy production were found: One may see it as optimizing the spatial organization of the convection cells, similar to the explicit simulation of e.g. wind circulation in the study of Kleidon et al. (2003).

A study that is linked to the Carnot limit (a system with fixed gradient where the limit is achieved when there are no frictional losses) is that of Konings et al. (2012) describing the hydrological cycle as a modified Rankine cycle. Their idealized atmosphere reached a maximum Carnot efficiency at a residual fraction of vapor in the atmosphere of 0.4, while the real water





cycles operates at values ranging between 0.5 to 0.7. The authors thus conclude that the system does not work at the Carnot limit. As this study uses the Carnot limit, it neglects feedbacks between fluxes and gradients. It could therefore be that this missing feedback is the cause for the mismatch between observed and optimized residual fraction.

The last study we would like to mention here is the study of Mölg (2015) who applied the MEP principle to derive parameters of an atmosphere-glacier system and reported MEP predicted values for fresh snow albedo and density close to calibrated values. However, there seems to be a flaw in their definition of entropy production, which was given by the ratio of mean heat flux and temperature. However, this only resembles the entropy increase by the colder reservoir, ignoring the entropy decrease by the warmer reservoir (so ignoring the term $1/T_h$ in Eq. 3). It is not clear if the study would yield similar results when the entropy production rate would have been defined correctly.

## 4.2 Applications from information entropy

Maximum entropy production has also been derived from a maximum entropy formalism for information entropy similar to the study of Dewar (2005). This approach also uses the term MEP; just like many of the studies discussed in section 3 using thermodynamic description of entropy production. However, it is not clear if it is related to the production of thermodynamic entropy (see e.g. Wang and Bras, 2009).

Within the field of Earth sciences, this approach has been mainly used by a series of paper by Wang and Bras (2009, 2010, 2011) and Huang and Wang (2016), using this formalism to successfully derive the partitioning of the different surface heat fluxes.

But also the application of del Jesus et al. (2012) on the spatial organization of tree, shrub and grass species, originates from this formalism: They used the derivation of Dewar (2010) for MEP at the ecosystem scale, which is defined as equivalent to maximizing photosynthesis, while there was no feedback implemented between the flux (photosynthesis) and the driving gradient. Still, they reported good correspondence to observed patterns despite the lack of a flux-gradient feedback. In that sense, this approach can be compared with that of the study of Börsing et al. (2017), who also looked at spatial organization without explicitly incorporating a flux-gradient feedback.

Until now, it is unclear if and when the information theoretical approach and the thermodynamic approach are similar. It is therefore difficult to compare the outcomes of both approaches.

## 4.3 Observed entropy production of different sites

There are also a couple of papers, looking at the entropy budgets of observed energy fluxes between the ecosystem and the atmosphere. The main hypothesis of these studies was that ecosystems being developed to a higher state (i.e. higher biomass) produce more entropy. Holdaway et al. (2010) and Lin (2015) accepted this hypothesis, while Quijano and Lin (2015) rejected it based on their observations. Because these analyzes were only based on observations, it is not possible to check how far each system is from the optimal state. It is also worth noting that they summed the entropy production terms of all measured fluxes (including radiative fluxes), which is different from what was discussed by Ozawa et al. (2003) and Pascale et al. (2012a).





Interestingly, Brunsell et al. (2011) tested the sensitivity of vegetation fraction to all kind of entropy production terms within a land-surface model, and came to similar conclusion as Holdaway et al. (2010) and Lin (2015). However, they also analyzed observed data obtained from three different eddy-covariance towers and showed that these did not give such a clear correlation between entropy production and vegetation cover. They indicated that eddy covariance observations do not contain all necessary
data to determine the entropy budget, such as temperatures higher up in the atmosphere. This may be one of the reasons why Quijano and Lin (2015) came to different conclusions than Holdaway et al. (2010) and Lin (2015).

## 5   Lessons learned and open questions

As shown in section 3, there have been several successful applications of TOPs in Earth sciences, and then especially in the atmospheric sciences. But unfortunately there exists no recipe book on how to apply TOPs. Still, there are several issues that
can be learned from the reviewed applications. Here we focus on which quantity to optimize and on the minimum requirements needed to apply TOPs. This is followed by a couple of open (research) questions.

### 5.1   Which quantity to optimize?

As can be seen from Table 1, most studies maximize entropy production and only a few maximize power. However, it should be noted that in the case when heat fluxes are optimized, MEP and MP give slightly different results due to the fact that in the
expression for power (Eq. 4), the temperature difference is divided by the temperature of the hot reservoir ($T_h$). This implies that a flux only dissipates its energy at the end of its trajectory, which is in reality not the case. So it it may be the case that the difference between MEP and MP may become smaller when higher resolution models are used (i.e. the dissipated energy of a flux adds its energy to the next sub-reservoir, which will then slightly increase its temperature). However, this should still be tested.
But also if the focus is not on heat fluxes, most studies use MEP (e.g. Kleidon and Schymanski, 2008; Schymanski et al., 2010; Porada et al., 2011; Wang et al., 2015). Generally, they use the expression formulated by Kleidon and Schymanski (2008), who defined entropy production as a flux times a potential difference divided by temperature. In isothermal conditions (which was assumed in these studies), maximizing power is thus mathematically the same as MEP. Nevertheless, the power formulation makes it more transparent that the focus is on energy conversions where the energy conversion itself de-
pletes the gradient driving this conversion. In many Earth systems, all power is ultimately being dissipated, making the MP principle exactly the same as the maximum dissipation principle. Additionally, the MP principle makes it easier to understand which flux to optimize (as been discussed by Ozawa et al., 2003; Pascale et al., 2012a).

### 5.2   Minimizing or maximizing?

While most of the cited studies in this paper maximized their objective function, there have also been a few that minimized
theirs. This happened in the case of the minimum energy expenditure/minimum dissipation, which have been applied to obtain



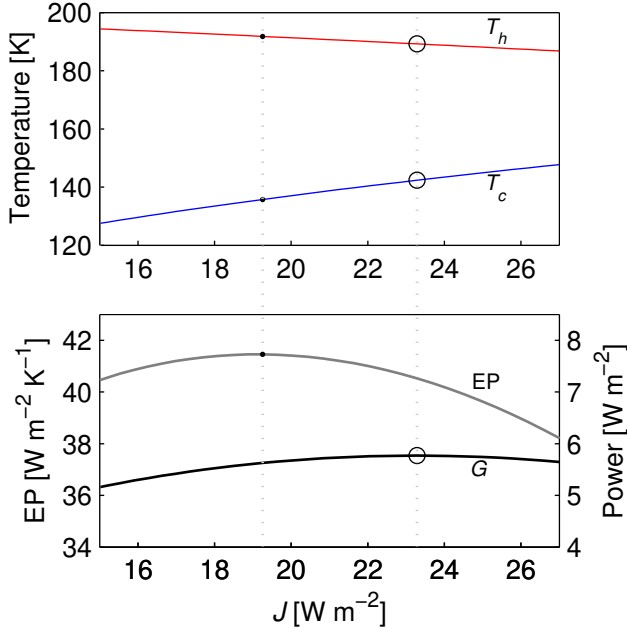

**Figure 6.** Difference between MEP and MP. The gray line denote the entropy production rate using the 2-box model of Fig. 1c and Eq. 3, while the black line denote power using the model of Fig. 1d and Eq. 4. The model is forced with $J_{in} = 96$ W m$^{-2}$ while $J_{h.out}$ and $J_{c.out}$ are given by the Stephan-Boltzman equation ($J_{x.out} = \sigma_{SB} T_x^4$, with $\sigma_{SB}$ being the Stephan-Boltzman constant). The difference in optimized $J$ is $J_{MP} - J_{MEP} \approx 4.0$ W m$^{-2}$, while the optimized temperatures differ 2.6 and 6.6 K for $T_h$ and $T_c$, respectively.

the spatial organization of flow networks (Leopold and Langbein, 1962; Rinaldo et al., 1992; Rodriguez-Iturbe et al., 1992a, b; Errera and Bejan, 1998; Hergarten et al., 2014). But also Paltridge (1975) used a minimization principle.

Although they seem to contradict the maximization principles, they are not: As discussed in section 3.4, minimum energy dissipation is in line with the maximum free energy dissipation principle proposed by Zehe et al. (2010, 2013), while
Kleidon et al. (2013) showed that dissipation being minimized within the network, is maximized outside the network. One can also look at it from another angle by questioning that if dissipation is minimized, then what happens to the incoming energy instead? For the case of the channel networks, a reasonable answer would be that this would go into the kinetic energy of water which can subsequently be used to e.g. erode river beds.

The minimization principle applied by Paltridge (1975) referred to the ratio of absorbed solar energy to outgoing longwave
radiation, which he later interpreted as a maximum in entropy production produced by the horizontal heat flux.

But besides these examples, the principle of minimum entropy production (Prigogine, 1947) applies to systems close to thermodynamic equilibrium (i.e. when driving gradients are absent). An example of this is a system of a connected hot and cold reservoir without energy transfer through its boundaries. Such a system evolves to a system where both reservoirs have equal temperature and the entropy production is zero. To distinguish between maximum and minimum entropy production, Reis
(2014) argued that minimum entropy production applies when the flux is fixed. This is the case in optimized flow networks,

where the amount of water flowing through the system is fixed by the effective precipitation, but also in the case of Fig. 1a, where the flux $J$ is fixed by $J_{in}$ (i.e. there is no 'leakage' or 'competing flux'). MEP applies according to Reis (2014) when the driving force (i.e. gradient) is fixed and the conductivities adapt themselves to reach their global maximum. However, this statement on MEP contradicts with the findings of most of the studies cited in this paper, where the flux (is required to) feed(s)

back on the gradient in order to obtain a maximum.

## 5.3  Minimum requirements

Although a clear recipe on how to use TOPs is missing, it is possible to define a list of minimum requirements. These are:

- There should be a flux-gradient feedback (even though the theoretical work of Reis, 2014, contradicts this statement, all applications described in this paper have this feedback). It should be noted that systems with fixed states boundary conditions, also find MEP derived states corresponding to observations. Some of them explicitly look at spatial organization of flows (e.g. del Jesus et al., 2012; Börsing et al., 2017), in which internal flux-gradients feedbacks are present.

- The system should have "enough" degrees of freedom to be able to evolve to the optimum state (note that "enough" is still fuzie at this stage. See also section 5.4): Jupp and Cox (2010) showed that not all systems are capable to do this.

- The system should be in (a quasi) steady state: Earth systems are hardly ever in true steady state. However, if boundary conditions change much slower than the process of interest, the steady state assumption may still be valid.

- The process descriptions in the model should be physically based: Westhoff and Zehe (2013) showed that parameters of a simple conceptual bucket model lacking the correct physics may give mathematical optima, but they are not physically meaningful.

Regarding the parameters that can potentially be optimized, we have seen in several papers that only parameters that in-

fluence the flux performing work (or has to do with motion) can be optimized (Ozawa et al., 2003; Pascale et al., 2012a; Westhoff and Zehe, 2013). In practice, this often means that resistances (or conductances) are the optimized parameters. Also the flux itself can be optimized directly.

Nevertheless, in the previous sections, we have discussed studies aiming to optimize different kinds of parameters. For example, Westhoff and Zehe (2013) concluded that, besides conductances, also a non-linearity factor could (mathematically)

be optimized, while Pascale et al. (2012a) did not discuss in detail what their optimized parameters really meant. Interestingly, both studies tried to optimize much more parameters, without considering a priori if it would make sense to optimize that specific parameter.

## 5.4  Open questions

Despite the significant amount of successful applications of TOPs, there are still quite a few open questions left. Here we

discuss several of them:





1. What is the effect of periodic forcing?

   Most studies assume a mean constant forcing bringing the system out of thermodynamic equilibrium. However, Westhoff et al. (2014) showed mathematically that periodic boundary conditions could lead to a second optimum[3]. This may not be of much concern in many of the atmospheric studies in which structures (eddies) are generally created relatively fast (com-

pared to the boundary conditions). But many other structures have been formed by extremes: e.g. extreme floods often shape the river structure abruptly, while the average river flow has much less influence. This aspect needs further investigation.

2. What are sufficient degrees of freedom?

   Jupp and Cox (2010) reflects on this with respect to planetary systems resulting in clear critical values separating systems

with and without sufficient degrees of freedom. But this analysis has not been conducted for other systems, making it unclear to which systems TOPs can be applied. Especially if TOPs are indeed inference principles, as suggested by several authors, then these degrees of freedom link to the question what the "essential physics" of the system under consideration are.

3. Are biotic factors (indirectly) influenced by thermodynamic limits?

To obtain a maximum in e.g. power, it generally means that the effective resistance/conductance (e.g. the effective heat conductance for the meridional heat flux) should be able to adapt. This adaptation is often done by the medium through which the optimized flux flows. In atmospheric applications of TOPs, this medium is generally air and the adaptation happens by the development of eddies. However, in many other Earth sciences discussed in this paper, the medium is often the soil or the vegetation with different adaptation process: For example, in soils, the effective flow resistance can

be changed by erosion or weathering, but also by biotic factors, such as worm burrows (which is the adaptation strategy assumed by Zehe et al., 2010, 2013). Although Zehe et al. (2013) speculate on why earthworms seem to obey TOPs, consensus is still lacking (see also the different conclusion of studies on ecosystem development by Holdaway et al., 2010; Lin, 2015; Quijano and Lin, 2015). More research is needed to understand how far ecosystem development can be expected to follow TOPs or if biological organizing principles, e.g. natural selection of the fittest, are in disagreement

with TOPs.

4. Is there a physical bottom-up explanation why a system would evolve to a thermodynamic limit?

   An often heard critique on TOPs is that it is unclear why a system would evolve to such an optimum state. Attempts have been made to explain this: e.g. Dewar (2003, 2005, 2009) approached it from the point of statistical mechanics and information theory, stating that MEP is the most probable state of a system, while Kleidon (2016) looked at it from

a combination of positive and negative feedback loops within a system. But also here consensus is lacking and more research is needed.

---

[3] a note should be made here that even though they assumed periodic forcing, they maximized the temporal mean power. It is not clear if and when this mean power is the quantity to be optimized, or if only the power during the forcing period should be taken into account.





5. How to design controlled experiments to rigorously test TOPs?

   Although a clear theory on why a system evolves to a thermodynamic limit is definitely beneficial, we would like to point
   out that many of the studies discussed in section 3 show empirical evidence that a system evolves to a thermodynamic
   limit. But because all these findings could be 'coincidence' (e.g. Rodgers, 1976), we argue that controlled experiments

should be designed to rigorously test these principles along a range of boundary conditions.

## 6   Conclusions

Thermodynamic optimality principles have often been used in (Earth) sciences to estimate model parameters or fluxes. How-
ever, it is not always clear what has to be optimized and how. In this paper we aimed to clarify terminology used in literature
and to infer on how these principles have been used and when they give proper predictions of observed fluxes and states.

In this review we distinguished between TOP applications with and without a flux-gradient feedback. Applications with such
a feedback have reported considerable success in predicting observed fluxes. Most of these successful applications used (exten-
sions of) one of the simple 2-box models shown in Fig. 1c-f, while optimizing the flux (or its effective conductance/resistance)
between the two boxes. The first applications all focused on temperature differences as the gradient driving the flux, but slowly
more and more applications with other kinds of energy conversions have been published. It also became clear that MEP is the

most widely used principle. But several studies showed that only fluxes that involve motion can be optimized (e.g. Ozawa et al.,
2001; Pascale et al., 2012a; Westhoff and Zehe, 2013). Therefore, the MP principle (which is generally the same as maximum
dissipation) is more precise on which fluxes to optimize. Especially for the kinds of conversions which are not driven by a
temperature difference, maximizing power or dissipation makes more sense than MEP.

   The criticism of e.g. Goody (2007) that many of the simple models lack important processes to correctly predict fluxes

with TOPs has appeared to be partly correct: There may be physical constraints preventing adaptation to such an optimal state
(or there are "essential physics" missing Dewar, 2009), but in many of the investigated systems Goody (2007) particularly
criticized, this was not the case (Jupp and Cox, 2010).

   Nevertheless, there are also several less convincing applications: in these applications fluxes are not described with the cor-
rect physics (Westhoff and Zehe, 2013; Westhoff et al., 2016; Wang et al., 2015), they do not correspond to observations(Porada et al.,

2011), or there seems to be some flaws in how the principle is applied (Leopold and Langbein, 1962; Mölg, 2015; Konings et al.,
2012).

   From all these applications, we filtered the following minimum set of requirements to apply TOPs: The system should be
in a (quasi) steady state; there must be a flux-gradient feedback; the system should have enough degrees of freedom; the flux
descriptions should be physically based; and only fluxes producing power can be optimized.

While in the above mentioned applications the flux-gradient feedback is the main reason for optima to exist, there have also
been applications without such a feedback. Among these, there are a number of MEP studies arising from information theory.
Although they show convincing predictions, it is still under debate if physical entropy production is inherent to information
theoretical entropy production.



Others used a fixed driving gradient and still showed observed fluxes or patterns in accordance with MEP. Generally, in these applications the flux-gradient feedback happens internally, where fluxes have been organized in high resolution models (del Jesus et al., 2012; Börsing et al., 2017).

The last group we distinguished are studies comparing entropy production of sites with different vegetation succession stages

to see if sites with a denser vegetation cover produces more entropy. However, conclusions of the different studies contradict each other, while they compared the summed entropy production of all terms of the energy balance with each other, which is not in line with the statement that only fluxes producing power can be optimized.

Despite the successful applications there remain critics on TOPS. Unless there is consensus on a theory explaining when and why natural processes tend to evolve according to (one of) the thermodynamic optimality principles, it will be hard to prove or

disprove the concept. We therefore think that with a more systematic (and transparent) MP approach the positive (or negative!) results will speak for themselves (a point also made by Blythe, 2016). But this is only possible if the principle is applied in a transparent and correct way.

*Author contributions.* MW outlined and wrote the manuscript, AK and EZ helped shaping the document from an early stage on. All other authors gave their input during a 3-day workshop hold in November 2016 in Amsterdam and they reviewed 3 to 5 papers that have been

discussed in this manuscript. They also proof-read the entire manuscript

*Competing interests.* The authors declare that they have no conflict of interest.

*Acknowledgements.* We would like to thank the Netherlands Earth System Science Center (NESSC) for their financial support (work201691) to host the 3-day workshop that formed the basis for this paper. SS is supported by the Luxembourg National Research Fund (FNR) AT-TRACT programme (A16/SR/11254288)



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
