# Peer review of "ESD Reviews: Thermodynamic optimality in Earth sciences. The missing constraints in modeling Earth system dynamics?"

_Earth System Dynamics, 2019_

## Referee Comment (RC1) · Michael Roderick (Referee) · 19 Mar 2019

**Review of Earth System Dynamics (ESD) Manuscript #2019-6**

Title:   Thermodynamic optimality in Earth sciences. The missing constraints in modeling Earth system dynamics?

Author: Westhoff et al

**Review**

The manuscript describes recent experience in using thermodynamic optimality principles (TOP) in the earth sciences.

The manuscript asserts that while TOP has been applied previously, that there has been much scepticism about the application of TOP because of [quoted from p. 3, lines 7-14]:

"*1) inconsistencies in the use of thermodynamic concepts and terminology; 2) the lack of a theory postulating a priori the respective ranges of applicability of the different TOPs Martyushev and Seleznev (2014) identified this issue as one of the main reasons for criticism on TOPs – and listed several restrictions of the maximum entropy production principle; 3) apparently arbitrary choices of processes and boundaries considered in systems with many interlinked processes producing power and entropy (Volk and Pauluis, 2010). 4) a lack of a widely accepted physical foundation that explains why a system should evolve to a maximum/minimum; and 5) claims that the MEP principle is an inference principle, making it a hypothesis that cannot be rejected (because you can always argue that a rejection is caused by a missing constraint: see e.g. Ross et al., 2012).*"

I agree with the authors that this is a reasonable summary of the current situation with the TOP field and the existing scepticism. However, on reading the article it was not clear what the overall approach was to address that scepticism.

On the one hand, the article begins with section 2 that describes the foundations of the topic. Figure 1 (but see below) was a potentially good start as was the preparation of Table 1 listing a number of previous studies and highlighting the foundations of those studies. However, on reading later sections (2, 3, 4), the level of synthesis did not emerge (for me at least) and these later parts tended to focus on a comprehensive description of what had been done. The description is too comprehensive and to this reader, I ended up being just as confused as when I started reading the manuscript.

In that respect, given the previous criticisms (e.g. Goody 2007, Ross et al 2012, J of Physical Chemistry, and others) I do not think that the current manuscript has adequately addressed its first principle aim of clarifying terminology and concepts.

To me at least, the problem here is that the authors have to decide whether (i) to produce a comprehensive list of all previous applications, or (ii) attempt to synthesise just a few of the previous studies into a coherent framework. I would strongly advocate for the latter option (ii) since I do not see the value of another comprehensive review given the previous high profile criticisms/scepticisms (e.g. Goody 2007, Ross et al 2012). On the latter synthesis approach advocated above, what is critical here is to have a strong foundation of language and formalism and to use that foundation to re-examine just a few key (previous) applications and attempt to cast those into a common intellectual framework. In particular, I was looking to Figure 1 for that formalism but after repeatedly reading the relevant text and studying Fig. 1, I realised that I did not understand either the figure or the related text.

To see my difficulty, let us start with Fig. 1a and compare that with Fig. 1b. In Fig. 1a we have three heat fluxes **Jin**, **J** (connecting the two boxes) and **Jout**. I originally thought that all would presumably have units of J (i.e., Joules) as those are the conventional units for heat (and work and energy change). However, there is no work shown on figure 1a (or on figure 1b). Now go to Figure 1b. We have a finite gap where "power" is ?done? (instead of work done) and this power is then dissipated to heat that is re-absorbed and leaves the entire systems+surroundings at a fixed temperature. While not given, the units for the power term (denoted with symbol G) are presumably J $s^{-1}$, i.e., power is the rate of doing work and so it has a time component. I imagine the dissipation (symbol used = D) must have the same units as those for G (J $s^{-1}$). But those units are different from the heat fluxes (which all have units of J and not J $s^{-1}$). Nowhere does either figure state where the work was done? Critically work must be done at the boundary and any work transfer is not associated with entropy change.

The same problem applies to the other two panels (Fig. 1cd, 1ef) of Fig. 1.

(An alternative interpretation is that the units of heat are J $s^{-1}$, and the diagrams are dimensionally correct. But on that interpretation this is not necessarily a carnot cycle!)

Now compare that with the conventional thermodynamic basis that many readers will have learnt in their earlier studies. The carnot cycle is a cycle combining 4 separate but reversible processes, that take a system from a defined state through a progression of different states (i.e., through a quasi-static set of changes) before returning to the starting state. The reversible processes are accompanied by flows of heat and work and when summed over the entire cycle they perform the maximum work possible such that any other sequence of states that return to the system to the start will have done less work. Of course the total work done plus the heat absorbed must equal zero around the cycle – that is how it is defined. Again, as I am sure the authors well know, that is all very interesting but the reversible processes are done in an infinitely long amount of time so this maximum work is not actually available and not very useful. Again, as the authors well know, any real process will be irreversible (e.g. friction examples given) and will produce less than the maximum possible work (of a reversible process) but the irreversible process can be completed in finite time. Hence in any real process we have to consider time if we want to consider irreversibility. Further, for an irreversible process, the change in entropy of the system plus surroundings must be greater than zero. However, this does not automatically mean that the entropy of the system will change in a particular way – for example, if heat flowed out of a closed system then the entropy will decrease (while the entropy of the surroundings increases). Importantly, entropy is associated with heat transfers but not with work transfers. This is the basic understanding that many potential readers will begin with. My suggestion is that if you want to start with a carnot cycle then you have to relate what you are trying to do to this classical understanding of a carnot cycle. Here I am talking about a series of quasi-static processes with constant mass over one cycle. Without establishing that underlying link to the readership, you risk losing (and ultimately confusing) people (like me) even more. What is the point of that?

The key point here is that the change in entropy during a reversible process becomes the reference against which the actual change in entropy in a real irreversible process can be compared. The entropy beyond that for a reversible process is the entropy related to irreversibility that is the foundation of this topic. The best explanation I have read on this topic is in the advanced graduate level textbook by A. Bejan (1997, Advanced Engineering Thermodynamics, $2^{nd}$ ed., John Wiley and Sons). I attach a photograph of p.110-111 that deals with several key points:

[Figure]

Figure. Photo of p110-111 in Bejan (1997).

Hopefully the scanned page is clear enough to be read.

Now this formulism did not begin with Bejan – it was established much earlier and can trace its roots back to the first paper by Willard Gibbs (in 1873) that introduced the temperature-entropy diagram (as an alternative to the pressure-volume diagram. (Although the ideas are old, the presentation of those ideas by Bejan is the best description that I have read.)

With a foundation like that it should be possible to start with a set of diagrams (energy, entropy) for closed and open systems, and then go through a few selected applications and identify the relevant terms (and ideally identify those terms which do not matter for the application in question).

In fact the above-noted framework can, and has been used for problems like flow with friction (Bejan, 1997, p. 136-138) or mixing (Bejan 1997, p. 138-141) or heat transfer across a finite gap (Bejan, 1997, p. 134-136) and these are all applications described in the current manuscript.

In summary, my overall suggestion is to change the basic approach of the manuscript. Instead of listing all the previous studies and approaches, etc., just take a few key studies and cast those in a consistent framework that allows people to see the link back to the carnot cycle

idea you began with while reformulating the carnot cycle to conventional understanding. With that approach you will address the first aim, i.e., about the inconsistency of thermodynamic concepts and terminology (p.3, lines 7-8).

I had many other comments but given my suggestion of a complete re-write then I did not give all details. The ones below were a few that I note here.

**Other Comments**

1.	p. 2, line 16, "In the final steady state …."  Do you mean at rest ? If so then please say it like that.

2.	P. 2, line 17. Asset ?? What does this mean? Also, in the same sentence you can remove the full stop, i.e., … conversions, and such limits ……

3.	P. 3, line 19. TYPO. several.

4.	P. 8, footnote. I did not understand the point of the footnote? Why not incorporate the text of the footnote into the main text.

5.	P. 14, line 10. ?TYPO? … physics **that** are required

6.	P. 15, line 4. ?distinguished? I think you might mean distinctive here.

7.	P. 15, lines 20-22. I did not understand this on several levels. Work transfers are not associated with entropy production. So if you minimise work then why does that also minimise entropy production? (Intuitively, if work is zero then all of the energy change is due to heat flow (in a closed system) and there is a maximum possibility of generating entropy. Further, with that objection aside, how is all this realised by work being performed uniformly along the channel.

8.	P. 20, line 30-31. I did not understand why a study based on observations means that it was not possible to assess how far from an optimal state the study was.

9.	P. 22, lines 12-15. I did not understand this example. If the system cannot transfer energy then it cannot transfer heat or do work. Why would this system evolve to a state of equal temperature in the absence of heat and work transfers?

10.	P. 25, lines 27-30. I did not understand this point. Power is the rate of doing work. Why is any optimisation restricted to those fluxes?

---

## Referee Comment (RC2) · Maarten Ambaum (Referee) · 29 Mar 2019

This paper is written by a broad range of authors with a high level of expertise, who are well placed to provide an overview of thermodynamic optimality principles in earth system science. It is a well written and very readable paper of appropriate length, although I did feel that at points the paper suffers from a lack of detail, so that any reader who is not completely familiar with the background material does need to refer back to the primary literature on all topics. I personally enjoyed reading the paper, and enjoyed picking up on some of the topics that I was less familiar with.

[Figure]

Reviews on this area have been written before, and I was looking at what this review adds above our present understanding; the paper promises quite a lot from the outset. I am afraid to say that in my opinion the paper does not deliver on these promises. In summary, the paper restates our lack of understanding about general properties of TOPs, but does not add any new insight.

The first sentence does not at all reflect my understanding of the area: thermodynamic optimality principles, as addressed in the present paper, have little solid physical foundation. My reading of the literature, including some of my own contributions, is that many people attempt to demonstrate an underlying optimality principle in some given system; that is different from people using optimality principles to estimate model parameters or fluxes that are not already known or estimated from other physically explicit models. These optimality principles are invariably used post-hoc, and not as calculating principles. Of course, the authors admit as much in the final paragraphs of the introduction.

In line 18 of the abstract we are promised that there is a correct and consistent use of the maximum power principle, which sounds far fetched to me. Maximum power is at best a hypothesis with a good amount of circumstantial evidence. However, it does not have the status of a physical theory which has a well defined application area and procedure. At best we can hope to give a geography of cases where an application of maximum power appears to give a physically realistic result. Of course, any new results or understanding in the rest of the paper could prove me wrong, but I do not think such new results or understanding were provided; in fact the paper mostly is a descriptive geography of applications of TOPs. Fun to read, but probably not adding much insight.

Your third paragraph (p.2, l. 12-20) is a case in point: this is a great example of a falling object reaching terminal velocity in the presence of friction. It is used to point to the potential of thermodynamic optimality principles. Of course, the ultimate balance at terminal velocity is between production of heat at the expense of potential energy, and

that seems to point to some possible thermodynamically optimal limit. But this does ignore the fundamental fact that, especially at higher Reynolds numbers, the primary, and limiting conversion is between potential and kinetic energy of the surrounding fluid. The balance is dynamic, not thermodynamic, evidenced by the fact that the energy conversion rate does not actually depend on the viscosity of the fluid, as long as the Reynolds number is large. The heat production is incidental; the terminal velocity is determined by a drag coefficient which is essentially a geometric property of the falling object.

Section 2.1: all basic and correct, but in section 2.2, and the rest I am surprised that there is never a reference to the Curzon-Ahlborn work on maximum power production which seems to me to address many issues of how heat is fluxed though a system using explicit models of conductive heat flux. The discussion around Eqs 3 and 4 imply a very particular physical set-up which is not at all explained in detail. For example: which flux is meant? The flux at the input terminal is typically different from that at the output terminal if mechanical work can be extracted. If the mechanical work is re-injected in the system, then we need to know at what temperature this happens. The schematic in Fig 1 implies this is re-injected at the output temperature (something that is not obvious at all: think about the thermodynamics around energy dissipation in tropical cyclones). I think you refer to this issue on p.21, l.15-21, but that was not particularly explicit. In other words, this part 2.2 does not explain much. Note also there are more explicit versions such as in Bejan's book and in my own book on Thermal Physics of the Atmosphere (Chapter 10.3) where you can find explicit expressions of "lost work" due to entropy production and of the effective temperatures of the input and output terminals. There you can also see that, for example, $T_c$ depends on where the heat is lost from the system, so it is a geometric property of the system as much as a thermodynamic property. In that sense, $T_c$ is not a function of J but of the whole of the fluid state. To take $T_c$ as a function of J is a statement of belief, or approximation, but not a statement of a physical principle.

[Figure]

Just as an aside to p.9, l. 31, as part of a PhD project (thesis by J. Kamieniecki, University of Reading, 2019) we repeated the work by Herbert et al. (2013) and found that they do not offer the complete picture in their paper: the profiles become really rather unrealistic above the levels they plot in their paper. So the interpretation of this result is that the MEP principle did produce profiles that look somewhat realistic in a limited part of the vertical column, but as a whole are unrealistic.

You refer to our work on p. 12, l. 19-27; the description is correct. The ultimate failure of our work has to do with the fact that energy conversions were dominated by latent heat fluxes which scale more with the mean temperature of the system, rather than the temperature gradients. My suspicion is that we need to somehow exclude chemical conversions, such as described by Pauluis in several papers as the "Gibbs penalty" (such as in Kamieniecki et al., 2018, J. Atmos. Sci.).

The authors are honest in that they do not avoid the fundamental failure of TOPs, such as in Section 3.3, which in my view should be interpreted as: several applications give broadly sensible results, but they often do not survive deeper scrutiny, or broadening of application range. It looks like the initial set-up of the physical problem encodes the outcome, not the TOP itself.

In section 3.4 I must admit that I am not an expert on this literature, but I thought that non-linear chemical reactions have been widely used to explain pattern formation in nature, with an essentially thermodynamic argument: the free energy being a non-linear function of some order parameter. Apologies if this sounds a bit vague, but I would have expected that a review, addressing pattern formation, would acknowledge that part of the physics literature which, as far as I understand it, is reasonably well established.

In section 3.5, I must again admit that this is not my specialist area of expertise, but the process described in p.17, l. 1 sounds very similar to the mechanism underlying the sandpile models of Per Bak, leading to SOC, which has a substantial body of literature
around it. I may well be completely wrong here, but I would be surprised if there was no link between the SOC states and some appropriate TOP state in such models. At the end of Section 3.5, the authors, admirably, point again to an observed limitation of TOPs.

In Sec. 5.1 you discuss MP vs. MEP, and essentially argue they point in broadly the same direction. I agree with that, but it also means that, in the absence of a first principles physical theory for TOPs, we cannot decide at this stage which of the two options is the better one. It is clear that MEP has the potential to be related to MaxEnt, while MP might well be related to ideas around stationary action principles in Lagrangian or Hamiltonian descriptions of physical systems. There does not seem to be an overriding argument presented either way.

Your discussion of minimum entropy production, as in Prigogine's work, seems to miss some clarity. I would have thought that minimum entropy production is a well established outcome for a system in the presence of linear fluxes (such as in Fourier's law) and fixed boundary conditions. The big transition to the kind of systems we are studying must then be the non-linearity of the fluxes, where flux values are linked to gradients in a non-trivial way, or the boundary conditions.

Your minimum requirements in section 5.3 appear mostly self-evident extractions from past experience, and therefore would not and should not exclude other ways that TOPs might be operational in the future.

Your set of questions in section 5.4 address some of the fundamental issues, especially questions 2 and 4; these are issues that everybody working in the field has been aware of for many years; it looks like this review has mostly restated these issues, and by giving a list of applications with successes and failures has only restated the gaps in our understanding.

Your very final sentence then summarizes what I dislike about this review: "But this is only possible of the principle is applied in a transparent and correct way". This

sentence implies there is such a thing as "the" principle, and that there is such a thing as the "correct" way. Both of these are highly disputed in the cited literature and the present review does not provide evidence for a solution for either of these.

I found the paper well written, and have therefore not picked up on any typos. Here are three I did spot and did record: p. 2, l. 21: an –> a p. 20, l. 27: "a couple" means 2; you mention 6. p. 23, l. 13: fuzie –> fuzzy

––––––––––––––––––––––––

---

## Author Comment (AC1) · 24 May 2019

We would like to thank Michael Roderick [MR] for his critical but constructive review. Below we will respond to issues raised.

[MR]The manuscript describes recent experience in using thermodynamic optimality principles (TOP) in the earth sciences.

The manuscript asserts that while TOP has been applied previously, that there has been much scepticism about the application of TOP because of [quoted from p. 3, lines 7-14]:

"*1) inconsistencies in the use of thermodynamic concepts and terminology; 2) the lack of a theory postulating a priori the respective ranges of applicability of the different TOPs Martyushev and Seleznev (2014) identified this issue as one of the main reasons for criticism on TOPs – and listed several restrictions of the maximum entropy production principle; 3) apparently arbitrary choices of processes and boundaries considered in systems with many interlinked processes producing power and entropy (Volk and Pauluis, 2010). 4) a lack of a widely accepted physical foundation that explains why a system should evolve to a maximum/minimum; and 5) claims that the MEP principle is an inference principle, making it a hypothesis that cannot be rejected (because you can always argue that a rejection is caused by a missing constraint: see e.g. Ross et al., 2012).*"

I agree with the authors that this is a reasonable summary of the current situation with the TOP field and the existing scepticism. However, on reading the article it was not clear what the overall approach was to address that scepticism.

We are glad that MR agrees with our summary of the situation. Our intentions to address the scepticism were on the one hand, to search for commonalities between different successful applications and to see what makes them successful. On the other hand, we aimed to clarify ambiguities around the term "maximum entropy production" and its use in many different contexts, sometimes leading to excessive room for interpretation and sometimes even to misleading results. By categorizing different approaches using a common two-box conceptual model, we aimed to make clear that not all studies should be interpreted in the same way. In the revised version, we will synthesize this more clearly.

[MR]On the one hand, the article begins with section 2 that describes the foundations of the topic. Figure 1 (but see below) was a potentially good start as was the preparation of Table 1 listing a number of previous studies and highlighting the foundations of those studies. However, on reading later sections (2, 3, 4), the level of synthesis did not emerge (for me at least) and these later parts tended to focus on a comprehensive description of what had been done. The description is too comprehensive and to this reader, I ended up being just as confused as when I started reading the manuscript.

In that respect, given the previous criticisms (e.g. Goody 2007, Ross et al 2012, J of Physical Chemistry, and others) I do not think that the current manuscript has adequately addressed its first principle aim of clarifying terminology and concepts.

Figure 1 is indeed the anchor point of the synthesis. In the revised manuscript, we will provide explicit exemplary calculations for each of the 6 cases, and synthesize the results more clearly. We will also focus the descriptions of the different studies more strongly to emphasize their relevance for Fig. 1.

[MR]To me at least, the problem here is that the authors have to decide whether (i) to produce a comprehensive list of all previous applications, or (ii) attempt to synthesise just a few of the previous studies into a coherent framework. I would strongly advocate for the latter option (ii) since I do not see the value of another comprehensive review given the previous high profile criticisms/scepticisms (e.g. Goody 2007, Ross et al 2012). On the latter synthesis approach advocated above, what is critical here is to have a strong foundation of language and formalism and to use that foundation to re-examine just a few key (previous) applications and attempt to cast those into a common intellectual framework.

We agree with MR to focus on Option ii) and choosing a few key examples to clarify concepts, but we also aim to demonstrate the generality of our arguments by putting all the relevant studies we could find into our concepts of explicit control volumes and energy and entropy budgets.

[MR] In particular, I was looking to Figure 1 for that formalism but after repeatedly reading the relevant text and studying Fig. 1, I realised that I did not understand either the figure or the related text.

To see my difficulty, let us start with Fig. 1a and compare that with Fig. 1b. In Fig. 1a we have three heat fluxes **Jin**, **J** (connecting the two boxes) and **Jout**. I originally thought that all would presumably have units of J (i.e., Joules) as those are the conventional units for heat (and work and energy change). However, there is no work shown on figure 1a (or on figure 1b). Now go to Figure 1b. We have a finite gap where "power" is ?done? (instead of work done) and this power is then dissipated to heat that is re-absorbed and leaves the entire systems+surroundings at a fixed temperature. While not given, the units for the power term (denoted with symbol G) are presumably J s-1, i.e., power is the rate of doing work and so it has a time component. I imagine the dissipation (symbol used = D) must have the same units as those for G (J s-1). But those units are different from the heat fluxes (which all have units of J and not J s-1). Nowhere does either figure state where the work was done? Critically work must be done at the boundary and any work transfer is not associated with entropy change.
The same problem applies to the other two panels (Fig. 1cd, 1ef) of Fig. 1.

(An alternative interpretation is that the units of heat are J s-1, and the diagrams are dimensionally correct. But on that interpretation this is not necessarily a carnot cycle!)

We apparently failed to explain this figure in a proper way, and we are thankful to MR for pointing out the gaps in our explanations. First of all, the units of all energy fluxes are J s$^{-1}$. A good point made by MR is that work (or power) is performed on the boundaries of a system. In the revised figure shown below (Fig. 1rev), we do include an explicit boundary around the two control volumes, clarifying that extracted power (G) refers to the work done on the boundaries and amounts to an extraction of energy (not entropy) from the system. We also clarify here that the heat produced by dissipation of the extracted work (D) has to enter across the same boundary across which power was extracted; hence it does not contribute to the internal entropy production. In this sense, Fig. 1 in the original manuscript was misleading. We will also accompany the revised version with an analysis of the energy and entropy balances of the general system in a Jupyter notebook, where all variables are explained with their units and where all necessary computations will be entirely transparent.

[Figure]

*Figure 1rev: Revised 2-box model with J being heat fluxes [W], G is power [W], D is dissipation [W] and T is temperature [K]. The first letter of the subscript refers to the origin of the flux, while the second one refers to the destination of the flux: o is outside, h is hot, c is cold.*

**[MR]** Now compare that with the conventional thermodynamic basis that many readers will have learnt in their earlier studies. The carnot cycle is a cycle combining 4 separate but reversible processes, that take a system from a defined state through a progression of different states (i.e., through a quasi-static set of changes) before returning to the starting state. The reversible processes are accompanied by flows of heat

and work and when summed over the entire cycle they perform the maximum work possible such that any other sequence of states that return to the system to the start will have done less work. Of course the total work done plus the heat absorbed must equal zero around the cycle – that is how it is defined. Again, as I am sure the authors well know, that is all very interesting but the reversible processes are done in an infinitely long amount of time so this maximum work is not actually available and not very useful. Again, as the authors well know, any real process will be irreversible (e.g. friction examples given) and will produce less than the maximum possible work (of a reversible process) but the irreversible process can be completed in finite time. Hence in any real process we have to consider time if we want to consider irreversibility. Further, for an irreversible process, the change in entropy of the system plus surroundings must be greater than zero. However, this does not automatically mean that the entropy of the system will change in a particular way – for example, if heat flowed out of a closed system then the entropy will decrease (while the entropy of the surroundings increases). Importantly, entropy is associated with heat transfers but not with work transfers. This is the basic understanding that many potential readers will begin with. My suggestion is that if you want to start with a carnot cycle then you have to relate what you are trying to do to this classical understanding of a carnot cycle. Here I am talking about a series of quasi-static processes with constant mass over one cycle. Without establishing that underlying link to the readership, you risk losing (and ultimately confusing) people (like me) even more. What is the point of that?

Thank you for raising these points. It is true that in thermodynamic textbooks the Carnot limit is typically derived from the Carnot cycle. And since this has been described in several textbooks (in the revised version we will explicitly refer to it), we will refer to the above time issue raised by MR and point out that in the our cases we do not consider the dynamics but steady-state conditions, assuming that the power extracted is limited by the above considerations.
Keeping the steady-state assumption in mind, the Carnot limit can also be derived in a different way: a heat flux between two connected reservoirs (as in Fig. 1a) has to produce entropy (eq. 1). If subsequently power is subtracted from the heat flux between the two reservoirs, then $J_{out} = J_{in} - G$. If this is substituted in Eq. 1 and solved for G (with sigma $= 0$), you end up with the Carnot limit (Eq. 2). As mentioned above, we identified some inconsistencies in our concepts and re-derived all equations using a common general setup, as presented in the above-mentioned Jupyter notebook, which will be made openly available and will be included as supplementary information.

**[MR]** The key point here is that the change in entropy during a reversible process becomes the reference against which the actual change in entropy in a real irreversible process can be compared. The entropy beyond that for a reversible process is the entropy related to irreversibility that is the foundation of this topic. The best explanation I have read on this topic is in the advanced graduate level textbook by A. Bejan (1997, Advanced Engineering Thermodynamics, 2nd ed., John Wiley and Sons). I attach a photograph of p.110-111 that deals with several key points:

Now this formulism did not begin with Bejan – it was established much earlier and can trace its roots back to the first paper by Willard Gibbs (in 1873) that introduced the temperature-entropy diagram (as an alternative to the pressure-volume diagram. (Although the ideas are old, the presentation of those ideas by Bejan is the best description that I have read.)

Thank you for this reference. Actually, in the equivalent section of the fourth edition (2016), Bejan explicitly refers to work done by the atmospheric system (p. 96), which is very relevant for many of the studies reviewed in our paper. We will refer to this section and the importance of considering non-steady-state conditions.

**[MR]** With a foundation like that it should be possible to start with a set of diagrams (energy, entropy) for closed and open systems, and then go through a few selected applications and identify the relevant terms (and ideally identify those terms which do not matter for the application in question).

We agree that it would be useful to extend our analysis to open, non-steady-state systems. For the sake of simplicity and for illustrating the general concept, we chose the simplest possible system expressing maxima in power and/or entropy production at certain conditions, i.e. a closed system

consisting of two closed reservoirs at steady state, only exchanging heat and entropy between each other and with the surroundings. For the extraction of power, we could have considered a heat engine between the two reservoirs, but we believe that it would not change the results and does not add to the clarity of the presentation.

**[MR]** In fact the above-noted framework can, and has been used for problems like flow with friction (Bejan, 1997, p. 136-138) or mixing (Bejan 1997, p. 138-141) or heat transfer across a finite gap (Bejan, 1997, p. 134-136) and these are all applications described in the current manuscript.

The aim of our analysis is to illustrate under which conditions extremes in extractable power and/or entropy production arise and can or have been used to make useful predictions about a natural system. For simplicity, we have chosen to start from the perspective of a system with fixed states (and thus from the Carnot limit), as this is generally well known by everybody with a basic thermodynamics background. We believe that the steps to more complex cases with flux-gradient feedbacks will be easier to grasp from this starting point. To better explain this in the revised manuscript, we will:
- Correct Fig 1, by adding a boundary around the coupled heat reservoirs and clarifying that extraction of power implies an export of energy but no entropy transport.
- Better explain that the Carnot limit is equivalent to the maximum power limit in the simplest case, and how it is linked to 0 entropy production by the flux conducting the work
- Add a more explicit discussion of the system variables that can be predicted using either the maximum entropy production or the maximum power principle.
- Accompany the manuscript with a Jupyter Notebook in which all the different fluxes present in Fig. 1 can be easily switched on or off to end up with the different configurations presented in Fig. 1

**[MR]** In summary, my overall suggestion is to change the basic approach of the manuscript. Instead of listing all the previous studies and approaches, etc., just take a few key studies and cast those in a consistent framework that allows people to see the link back to the carnot cycle idea you began with while reformulating the carnot cycle to conventional understanding. With that approach you will address the first aim, i.e., about the inconsistency of thermodynamic concepts and terminology (p.3, lines 7-8).

As explained above, we will take most of the suggestions of MR into account to better explain the step from the Carnot limit to the maximum power limit. We will explain successful approaches and common pitfalls using a small number of example studies, followed by a short classification of the many studies we reviewed into different system representations as outlined in Fig. 1.

**[MR]** I had many other comments but given my suggestion of a complete re-write then I did not give all details. The ones below were a few that I note here.

**Other Comments**
1. p. 2, line 16, "In the final steady state …." Do you mean at rest? If so then please say it like that.

Thank you for pointing out this opacity. What we meant as "final steady state" is that acceleration is 0 and velocity constant, i.e. the forces related to friction and gravity balance. This means that all the potential energy is dissipated as heat and none is transferred to the kinetic energy of the object any more. We will clarify this in the revised text.

2. P. 2, line 17. Asset ?? What does this mean? Also, in the same sentence you can remove the full stop, i.e., … conversions, and such limits ……

We will reformulate to: "Thermodynamics provides fundamental limits to such conversions and in some cases extremum conditions (maxima or minima in conversion rates) that can be used to predict system states."

3. P. 3, line 19. TYPO. several.

Thank you for pointing this out

4. P. 8, footnote. I did not understand the point of the footnote? Why not incorporate the text of the footnote into the main text.

     We used a footnote to deconvolute the sentence, but we realise that the sentence is still too convoluted. We will re-formulate Lines 1-11 to:

"It has been argued that MEP is not a physical but rather an inference principle, (e.g. Goody, 2007; Dewar, 2009), e.g. to obtain macroscopic properties of a system using the relevant physical constraints. Dewar (2009) explained that the MEP principle could be derived from maximum entropy (MaxEnt) formalism, which has its origin in information theory, and can be seen as a way to infer if a system is described by sufficient physical assumption and constraints – which Dewar called the "essential physics". It then depends on the system of interest what is the essential physics. With this respect, he argued that, given the close correspondence with observations, the simple MEP-based atmospheric model of Paltridge (1978, see section 3.1) apparently covers the essential physics, while all other physical details appear to be irrelevant at that scale. Dewar (2009) pointed out that previous attempts to derive MEP form the MaxEnt formalism (Dewar, 2003, 2005) had different flaws, but maintains that the two are likely connected. He concludes that *"If MEP is not a faithful expression of MaxEnt, the implication is that MEP itself introduces some additional physical assumptions beyond those explicitly identified in the problem (…), only when those additional assumptions are valid (or invalid but irrelevant) would MEP be practically useful as a Messenger of Essential Physics."* "

5. P. 14, line 10. ?TYPO? … physics **that** are required
     Yes, thank you for pointing this out

6. P. 15, line 4. ?distinguished? I think you might mean distinctive here.
     Yes, indeed. Thank you

7. P. 15, lines 20-22. I did not understand this on several levels. Work transfers are not associated with entropy production. So if you minimise work then why does that also minimise entropy production? (Intuitively, if work is zero then all of the energy change is due to heat flow (in a closed system) and there is a maximum possibility of generating entropy. Further, with that objection aside, how is all this realised by work being performed uniformly along the channel.

     This is indeed a very good point. They expressed entropy production as the amount of work divided by temperature, with the notion that in case of a mass flux they did not divide by the temperature, but by the geopotential.

The idea behind this is that all performed work is also dissipated inside the system by friction into heat. The rate of entropy production is thus proportional to the rate of (dissipated) work. We actually re-stated this assumption in the manuscript, but now found that this concept is flawed, as work is an energy flow out of the system without an associated entropy flow, as MR pointed out correctly. Therefore, heat generated by dissipation of that energy should not be considered part of the internal entropy production but part of the entropy exchange with the surroundings. We will explain this better in the revised manuscript.

8. P. 20, line 30-31. I did not understand why a study based on observations means that it was not possible to assess how far from an optimal state the study was.

     Since the studies did not present a theoretical optimal state, it is not possible to assess how far their systems were away from that. However, this is irrelevant for their hypotheses (i.e. more biomass leads to higher EP), so we will remove this statement.

9. P. 22, lines 12-15. I did not understand this example. If the system cannot transfer energy then it cannot transfer heat or do work. Why would this system evolve to a state of equal temperature in the absence of heat and work transfers?

     This example refers to an isolated system of two reservoirs with initially different temperatures. Since heat flows from the hot to the cold reservoir, the final state would be one in which the temperature

of both reservoirs are equal and heat transfer has ceased (thermodynamic equilibrium). However, the minimum entropy production principle actually applies to much less trivial circumstances, involving coupled fluxes approaching a non-equilibrium steady state (see Kondepudi & Prigogine, 1998, Capter 17.2). We will re-formulate the statement to:

"Besides these examples, systems close to thermodynamic equilibrium approaching steady state follow the principle of minimum entropy production (Prigogine, 1947). A trivial example of this is an isolated system of connected hot and cold reservoirs without energy transfer through its external boundaries. Such a system evolves to a system where both reservoirs have equal temperature and the entropy production is zero (thermodynamic equilibrium). More interesting examples are given by e.g. Kondepudi & Prigogine (1998, Chapter 17.2)."

10. P. 25, lines 27-30. I did not understand this point. Power is the rate of doing work. Why is any optimisation restricted to those fluxes?

Thank you for questioning this statement; there is actually more wrong with it.
We deduced it from the reviewed literature, where the 'successful' applications only optimized fluxes or parameters describing these fluxes, while attempts to optimize other fluxes failed.
We will re-formulate to:
"Based on all these applications and our own simple analysis, we conclude that the MEP and MP principles are potentially useful if:
- the system is in a (quasi) steady state;
- there are sufficient flux-gradient feedbacks to result in non-linearity in power or entropy production;
- the system has enough degrees of freedom;
- the flux descriptions are physically based (related to thermodynamic forces);
- a complete balance of entropy and the exchanged quantities can be computed;
- the system definition results in steady-state conditions which are NOT in thermodynamic equilibrium.
If the above conditions are met, it is possible (under certain conditions) to find an optimal value of a flux coefficient that would lead to a maximum in entropy production by this flux or a maximum in extractable power from this flux. These optima are not necessarily the same and it is a matter of additional experimental and theoretical investigation to decide a priori if one of them is the correct attractor for a given system."

With kind regards,

On behalf of all coauthors,

Martijn Westhoff

---

## Author Comment (AC2) · 24 May 2019

We would like to thank Maarten Ambaum (MA) for his critical, but constructive review. Below, we answer to all points raised.

[MA] This paper is written by a broad range of authors with a high level of expertise, who are well placed to provide an overview of thermodynamic optimality principles in earth system science. It is a well written and very readable paper of appropriate length, although I did feel that at points the paper suffers from a lack of detail, so that any reader who is not completely familiar with the background material does need to refer back to the primary literature on all topics. I personally enjoyed reading the paper, and enjoyed picking up on some of the topics that I was less familiar with.

Thank you for the kind words regarding readability of the manuscript. Regarding the lack of detail, we think it is always difficult to find that balance in a review paper. Reviewer 1, advised to be less comprehensive when describing the different studies. We will therefore critically look which literature needs to be described in more detail and which ones can do with less.

[MA] Reviews on this area have been written before, and I was looking at what this review adds above our present understanding; the paper promises quite a lot from the outset. I am afraid to say that in my opinion the paper does not deliver on these promises. In summary, the paper restates our lack of understanding about general properties of TOPs, but does not add any new insight.

Our aim was to find commonalities between successful applications in order to learn from them what is needed to apply TOPs. Subsequently, we aimed to clarify ambiguities around the term "maximum entropy production" and its use in many different contexts, sometimes leading to excessive room for interpretation and sometimes even to misleading results. By distinguishing between these different methods/forms we aimed to make clear that not all studies should be interpreted the same way. Based on input from Reviewer 1, we will focus in the revised version on choosing a few key examples to clarify concepts, but we also aim to demonstrate the generality of our arguments by putting all the relevant studies we could find into our concepts of explicit control volumes and energy and entropy budgets.

[MA] The first sentence does not at all reflect my understanding of the area: thermodynamic optimality principles, as addressed in the present paper, have little solid physical foundation. My reading of the literature, including some of my own contributions, is that many people attempt to demonstrate an underlying optimality principle in some given system; that is different from people using optimality principles to estimate model parameters or fluxes that are not already known or estimated from other physically explicit models. These optimality principles are invariably used post-hoc, and not as calculating principles. Of course, the authors admit as much in the final paragraphs of the introduction.

We guess that this comment refers to the first sentence of the abstract? We did not want to give the impression that TOPs have a solid physical foundation, since we agree with MA that they have not. The different ways TOPs are applied (demonstrating an underlying optimality principle vs estimating model parameters) are in our opinion supplementary to each other: The former one is needed to test if the principle can be used to get meaningful results, while the latter can only be done if one knows that the applied principle leads to meaningful results. We nevertheless agree with MA that in the literature often only one of the two ways is explored.

[MA] line 18 of the abstract we are promised that there is a correct and consistent use of the maximum power principle, which sounds farfetched to me. Maximum power is at best a hypothesis with a good amount of circumstantial evidence. However, it does not have the status of a physical theory which has a well-defined application area and procedure. At best we can hope to give a geography of cases where an application of maximum power appears to give a physically realistic

result. Of course, any new results or understanding in the rest of the paper could prove me wrong, but I do not think such new results or understanding were provided; in fact the paper mostly is a descriptive geography of applications of TOPs. Fun to read, but probably not adding much insight.

With hindsight, we agree that this claim in the abstract is indeed too strong, and we will change this in the revised version where we will put more emphasis on synthesizing the results. Regarding the "geography of cases…", we want to emphasize that it was exactly our aim to investigate these cases to find commonalities between successful applications vs applications using different methods (but same terminology). As explained above, we will focus in the revised version on choosing a few key examples to clarify the different concepts.

[MA] Your third paragraph (p.2, l. 12-20) is a case in point: this is a great example of a falling object reaching terminal velocity in the presence of friction. It is used to point to the potential of thermodynamic optimality principles. Of course, the ultimate balance at terminal velocity is between production of heat at the expense of potential energy, and that seems to point to some possible thermodynamically optimal limit. But this does ignore the fundamental fact that, especially at higher Reynolds numbers, the primary, and limiting conversion is between potential and kinetic energy of the surrounding fluid. The balance is dynamic, not thermodynamic, evidenced by the fact that the energy conversion rate does not actually depend on the viscosity of the fluid, as long as the Reynolds number is large. The heat production is incidental; the terminal velocity is determined by a drag coefficient which is essentially a geometric property of the falling object.

It is indeed a good point that the surrounding fluid also gets kinetic energy. However, this kinetic energy is subsequently dissipated into heat as well. We will add this to the revised version. However, we do not agree that the "balance is dynamic, not thermodynamic", since this assumes that thermodynamics only deals with heat. Thermodynamics deals with all kind of energies (including mechanical energies) as is expressed in the first law of thermodynamics. Besides that this example of a falling object should be seen more as an analogue for than an example of application of a thermodynamic principle

[MA] Section 2.1: all basic and correct, but in section 2.2, and the rest I am surprised that there is never a reference to the Curzon-Ahlborn work on maximum power production which seems to me to address many issues of how heat is fluxed though a system using explicit models of conductive heat flux.

The Curzon-Ahlborn work on maximum power production is an extension of the classical reversible thermodynamics (on which the Carnot limit is based) to irreversible thermodynamics. But it still deals with heat engines where the heat reservoirs operate at fixed temperatures. Since our focus is on studies with flux-gradient feedbacks, the Curzon-Ahlborn work on maximum power production is out of the scope of this work. On top of that, none of the reviewed literature in section 3 and 4 referred to this type of work, so we don't feel that this should be part of this review.

[MA] The discussion around Eqs 3 and 4 imply a very particular physical set-up which is not at all explained in detail. For example: which flux is meant? The flux at the input terminal is typically different from that at the output terminal if mechanical work can be extracted. If the mechanical work is re-injected in the system, then we need to know at what temperature this happens. The schematic in Fig 1 implies this is re-injected at the output temperature (something that is not obvious at all: think about the thermodynamics around energy dissipation in tropical cyclones). I think you refer to this issue on p.21, l.15-21, but that was not particularly explicit. In other words, this part 2.2 does not explain much. Note also there are more explicit versions such as in Bejan's book and in my own book on Thermal Physics of the Atmosphere (Chapter 10.3) where you can find explicit expressions of

"lost work" due to entropy production and of the effective temperatures of the input and output terminals. There you can also see that, for example, T_c depends on where the heat is lost from the system, so it is a geometric property of the system as much as a thermodynamic property. In that sense, T_c is not a function of J but of the whole of the fluid state. To take T_c as a function of J is a statement of belief, or approximation, but not a statement of a physical principle.

Thank you for pointing this out. Eq. 3 and 4 are connected to Fig. 1c-f, which we will explain better in the revised manuscript. The reason to start with the simple setups of Fig. 1 is to illustrate the general concept by choosing the simplest possible system expressing maxima in power and/or entropy production at certain conditions, i.e. a closed system consisting of two closed reservoirs at steady state, only exchanging heat and entropy between each other and with the surroundings. MA states correctly that $T_c$ is a function of more than only $J$, but for the sake of simplicity, we stick with this relation (also because in the reviewed literature this assumption is generally made). We will add a sentence to state that $T_c$ depends on more than $J$.

Please note that (as reviewer 1 pointed out) there were some flaws in Fig. 1, of which the major flaw is that work should be performed on the boundary of the system. Therefore, we will revise this figure: In the revised figure shown below (Fig. 1rev), we do include an explicit boundary around the two control volumes, clarifying that extracted power (G) refers to the work done on the boundaries and amounts to an extraction of energy (not entropy) from the system. We also clarify here that the heat produced by dissipation of the extracted work (D) has to enter across the same boundary across which power was extracted; hence it does not contribute to the internal entropy production. In this sense, Fig. 1 in the original manuscript was misleading. We will also accompany the revised version with an analysis of the energy and entropy balances of the general system in a Jupyter notebook, where all variables are explained with their units and where all necessary computations will be entirely transparent.

[Figure]

*Figure 1rev: Revised 2-box model with J being heat fluxes [W], G is power [W], D is dissipation [W] and T is temperature [K]. The first letter of the subscript refers to the origin of the flux, while the second one refers to the destination of the flux: o is outside, h is hot, c is cold.*

The most important message in section 2.2 is that we move from systems with fixed state boundaries (which are dealt with in classical textbook thermodynamics) to systems with fixed input rates. This is a crucial step to understand why (mathematical) optima exist, while also all literature reviewed in section 3 deals with this. We will stress this more clearly in the revised manuscript, while also explicitly referring to the differences with the classical textbook thermodynamics.

**[MA]** Just as an aside to p.9, l. 31, as part of a PhD project (thesis by J. Kamieniecki, University of Reading, 2019) we repeated the work by Herbert et al. (2013) and found that they do not offer the complete picture in their paper: the profiles become really rather unrealistic above the levels they plot in their paper. So the interpretation of this result is that the MEP principle did produce profiles that look somewhat realistic in a limited part of the vertical column, but as a whole are unrealistic.

Thank you for pointing this out. We will try to get hands on it to be able to refer to it.

**[MA]** You refer to our work on p. 12, l. 19-27; the description is correct. The ultimate failure of our work has to do with the fact that energy conversions were dominated by latent heat fluxes which scale more with the mean temperature of the system, rather than the temperature gradients. My suspicion is that we need to somehow exclude chemical conversions, such as described by Pauluis in several papers as the "Gibbs penalty" (such as in Kamieniecki et al., 2018, J. Atmos. Sci.).

Thank you for pointing this out. What we got out of the paper of Pascale et al. (2012a) was indeed that the material entropy production was dominated by the latent heat fluxes (i.e. chemical conversions), which do not feedback on the temperature gradient. So our explanation is in essence the same as MA is suggesting here. We will therefore stress in the revised version that latent heat fluxes are chemical conversions, which seem to be left out of the maximization.

**[MA]** The authors are honest in that they do not avoid the fundamental failure of TOPs, such as in Section 3.3, which in my view should be interpreted as: several applications give broadly sensible results, but they often do not survive deeper scrutiny, or broadening of application range. It looks like the initial set-up of the physical problem encodes the outcome, not the TOP itself.

We indeed wanted to be honest about it. However, we have to better include that in our synthesis.

**[MA]** In section 3.4 I must admit that I am not an expert on this literature, but I thought that non-linear chemical reactions have been widely used to explain pattern formation in nature, with an essentially thermodynamic argument: the free energy being a nonlinear function of some order parameter. Apologies if this sounds a bit vague, but I would have expected that a review, addressing pattern formation, would acknowledge that part of the physics literature which, as far as I understand it, is reasonably well established.

Thank you for pointing this out. We did not look into this part of the literature since our focus is on applications in Earth sciences. And since we will slightly change the focus in the revised manuscript to deriving the entropy balance for a set of cases we will not include this in the revised manuscript.

**[MA]** In section 3.5, I must again admit that this is not my specialist area of expertise, but the process described in p.17, l. 1 sounds very similar to the mechanism underlying the sandpile models of Per Bak, leading to SOC, which has a substantial body of literature around it. I may well be completely wrong here, but I would be surprised if there was no link between the SOC states and some appropriate TOP state in such models. At the end of Section 3.5, the authors, admirably, point again to an observed limitation of TOPs.

Thank you for the suggestion (MA probably refers to p16 (section 3.4)). A quick look told us that it has indeed some commonalities: especially the power law relations which are also present when describing some statistics regarding the derived Optimal Channel Networks (OCN). This will be made explicit in the revised version.

**[MA]** In Sec. 5.1 you discuss MP vs. MEP, and essentially argue they point in broadly the same direction. I agree with that, but it also means that, in the absence of a first principles physical theory

for TOPs, we cannot decide at this stage which of the two options is the better one. It is clear that MEP has the potential to be related to MaxEnt, while MP might well be related to ideas around stationary action principles in Lagrangian or Hamiltonian descriptions of physical systems. There does not seem to be an overriding argument presented either way.

       With hindsight, we agree that we could (and should) not have made this statement here.

**[MA]** Your discussion of minimum entropy production, as in Prigogine's work, seems to miss some clarity. I would have thought that minimum entropy production is a well established outcome for a system in the presence of linear fluxes (such as in Fourier's law) and fixed boundary conditions. The big transition to the kind of systems we are studying must then be the non-linearity of the fluxes, where flux values are linked to gradients in a non-trivial way, or the boundary conditions.

       This is indeed correct. The example described here refers to an isolated system of two reservoirs with initially different temperatures. Since heat flows from the hot to the cold reservoir, the final state would be one in which the temperature of both reservoirs are equal and heat transfer has ceased (thermodynamic equilibrium). However, the minimum entropy production principle actually applies to much less trivial circumstances, involving coupled fluxes approaching a non-equilibrium steady state (see Kondepudi & Prigogine, 1998, Chapter 17.2). We will re-formulate the P22, L11-14 to:
"Besides these examples, systems close to thermodynamic equilibrium approaching steady state follow the principle of minimum entropy production (Prigogine, 1947). A trivial example of this is an isolated system of connected hot and cold reservoirs without energy transfer through its boundaries. Such a system evolves to a system where both reservoirs have equal temperature and the entropy production is zero (thermodynamic equilibrium). More interesting examples are given by e.g. Kondepudi & Prigogine (1998, Chapter 17.2)."

**[MA]** Your minimum requirements in section 5.3 appear mostly self-evident extractions from past experience, and therefore would not and should not exclude other ways that TOPs might be operational in the future.

       Yes, we also realize that the way we described this is too firm. In fact it is an outcome of (or lessons learned form) the literature study. In the revised manuscript we will also formulate it as:
"Based on all these applications and our own simple analysis, we conclude that the MEP and MP principles are potentially useful if:
- the system is in a (quasi) steady state;
- there are sufficient flux-gradient feedbacks to result in non-linearity in power or entropy production;
- the system has enough degrees of freedom;
- the flux descriptions are physically based (related to thermodynamic forces);
- a complete balance of entropy and the exchanged quantities can be computed;
- the system definition results in non-trivial steady-state conditions.
If the above conditions are met, it is possible (under certain conditions) to find an optimal value of a flux coefficient that would lead to a maximum in entropy production by this flux or a maximum in extractable power from this flux. These optima are not necessarily the same and it is a matter of additional experimental and theoretical investigation to decide a priori if one of them is the correct attractor for a given system."

**[MA]** Your set of questions in section 5.4 address some of the fundamental issues, especially questions 2 and 4; these are issues that everybody working in the field has been aware of for many years; it looks like this review has mostly restated these issues, and by giving a list of applications with successes and failures has only restated the gaps in our understanding.

We are conscious that people well acquainted in this field are aware of these questions. Nevertheless, we think they are still useful to guide future research, while we also mention which studies are exemplary for that specific question, and which studies have already (partly) dealt with that question.

**[MA]** Your very final sentence then summarizes what I dislike about this review: "But this is only possible of the principle is applied in a transparent and correct way". This sentence implies there is such a thing as "the" principle, and that there is such a thing as the "correct" way. Both of these are highly disputed in the cited literature and the present review does not provide evidence for a solution for either of these.

We can see why you dislike it and, as said before, we agree that such a conclusion is too firm. Therefore we will state that the topic cannot finally be settled yet and that this is work in progress.

**[MA]** I found the paper well written, and have therefore not picked up on any typos. Here are three I did spot and did record: p. 2, l. 21: an –> a p. 20, l. 27: "a couple" means 2; you mention 6. p. 23, l. 13: fuzie –> fuzzy

Thank you for pointing this out. These will be corrected.

With kind regards,

On behalf of all coauthors,

Martijn Westhoff